# Stochastic Shortest Path with Sparse Adversarial Costs

Emmeran Johnson[1]*    Alberto Rumi[2,3]*    Ciara Pike-Burke[1]    Patrick Rebeschini[4]

[1] Department of Mathematics, Imperial College London
[2] Università degli Studi di Milano
[3] CENTAI Institute
[4] Department of Statistics, University of Oxford

## Abstract

We study the adversarial Stochastic Shortest Path (SSP) problem with sparse costs under full-information feedback. In the known transition setting, existing bounds based on Online Mirror Descent (OMD) with negative-entropy regularization scale with $\sqrt{\log SA}$, where $SA$ is the size of the state-action space. While we show that this is optimal in the worst-case, this bound fails to capture the benefits of sparsity when only a small number $M \ll SA$ of state-action pairs incur cost. In fact, we also show that the negative-entropy is inherently non-adaptive to sparsity: it provably incurs regret scaling with $\sqrt{\log S}$ on sparse problems. Instead, we propose a family of $\ell_r$-norm regularizers ($r \in (1,2)$) that adapts to the sparsity and achieves regret scaling with $\sqrt{\log M}$ instead of $\sqrt{\log SA}$. We show this is optimal via a matching lower bound, highlighting that $M$ captures the effective dimension of the problem instead of $SA$. Finally, in the unknown transition setting the benefits of sparsity are limited: we prove that even on sparse problems, the minimax regret for any learner scales polynomially with $SA$.

## 1 Introduction

The Stochastic Shortest Path (SSP) problem is a fundamental model in reinforcement learning [1, 32], which describes tasks where an agent interacts with an environment over episodes and must reach a designated goal state within each episode while minimizing accumulated costs. This covers problems such as car navigation while trying to avoid traffic jams, or internet routing. Recently, this classical setting has been extended to the adversarial regime, where costs may vary arbitrarily between episodes [28, 8, 6, 38] and the goal is to obtain theoretical guarantees robust to any cost-generation mechanism. Under *full-information* feedback where the full cost vector is observed after each episode and known transitions, current algorithms achieve regret bounds that scale as $\mathcal{O}\left(\sqrt{DKT_\star \log SAT_\star}\right)$, where $D$ is the diameter (the smallest expected hitting time of any policy from any state), $T_\star$ is the expected hitting time of the optimal policy, $S$ is the number of states, $A$ is the number of actions, and $K$ is the number of episodes. These bounds are independent of any cost structure and are shown to be minimax optimal up to logarithmic factors in [8].

In SSP, the size of the state-action space $SA$ – which we consider and refer to as the *dimension* of the problem – appears in the minimax regret as $\sqrt{\log SA}$. While in the worst-case this is unimprovable (we show this in Theorem 3.1), many real-world problems have costs with structural properties that may be leveraged for improved regret. A common property often considered in the statistics and machine learning literature [36] is sparsity, which can naturally arise for SSP problems. For instance, in the car navigation example, the number of traffic jams is usually much smaller than the number of roads. Motivated by this, we consider *sparse* SSP problems where $M$, the maximum number of state-action pairs with non-zero cost in an episode, can be much smaller than $SA$.

In such scenarios, the regret bounds should capture some dependence on $M$, reflecting an improvement in performance on easier sparse problems as $M \to 1$ and recovering the standard bounds on

---

*Equal contribution, corresponding author: `emmeran.johnson17@imperial.ac.uk`

worst-case problems as $M \to SA$. In fact, for the so-called experts setting[2] ($S = 1$), the minimax regret scales with $\sqrt{KMA^{-1} \log A}$ instead of $\sqrt{K \log A}$ in the worst-case [19], providing a polynomial improvement in the dimension $A$. Furthermore, this is achieved with Online Mirror Descent (OMD) with negative entropy regularization. However for SSP where the problem dimension also includes the size of the state space $S$, we show that existing approaches also based on OMD with the same negative entropy regularization [8] fail to exploit sparse costs. We construct a sparse SSP problem where this algorithm suffers $\sqrt{\log S}$ regret (Theorem 3.1), providing no improvement in terms of $S$ compared to the minimax regret for the non-sparse worst-case $M = SA$ problem. The failure of existing SSP methods to exploit sparsity, let alone match the polynomial improvements from the experts setting, leads us to ask the following questions:

*Does sparsity improve the minimax-regret in the full-information feedback SSP problem? How much?*

We answer the first question positively for the known-transition case by designing a family of regularizers based on $\ell_r$-norms for $r \in (1, 2)$ for which we show a $\mathcal{O}(\sqrt{DKT_\star \log MT_\star})$ regret bound that depends logarithmically on the sparsity level $M$, rather than on the size of the state-action space $SA$, without requiring the knowledge of $M$ in advance. This family of regularizers interpolates between the negative entropy and the squared Euclidean norm (see Section 4), allowing flexibility for much weaker regularization on sparse points in sparse settings and recovering existing algorithms (and guarantees) in the non-sparse setting.

We show that the above $\sqrt{\log M}$ dependence on $M$ is unimprovable by constructing a matching lower bound. Interestingly, this establishes that the benefit of sparsity in SSP is logarithmic in $SA$ instead of polynomial, as in the simpler experts problem, thus answering the second question. It also highlights that $M$ plays the role of effective dimension, replacing the general dimension $SA$ in controlling the scaling of the minimax regret.

While the benefits of sparsity in SSP are only logarithmic, we emphasize that due to the often combinatorial nature of the state-action space, these improvements can be significant. For instance, in many problems the size of the state-action space grows exponentially in some parameters, while the assigned costs remain linear or even constant. This occurs in many real-world problems (e.g. [31, 30]) in which settings exploiting sparsity can lead to polynomial improvements.

Finally, it is natural to ask whether sparsity may provide similar benefits in the unknown transition setting. However, in Theorem 5.1, we show a lower bound with polynomial dependence on $SA$ in a sparse SSP instance with unknown transitions. This illustrates that in the unknown transition setting the sparsity level $M$ does not play the same role of effective dimension, and that the general dimension $SA$ is crucial in controlling the scaling of the minimax regret polynomially, motivating our focus on the known transitions setting. In particular, this result shows that sparse problems with combinatorial state-action spaces will remain very challenging.

**Our results provide a complete characterisation of the benefits of sparsity in removing dimension dependence (i.e. $SA$) for adversarial SSP problems under full-information feedback.**

## 1.1 Contributions

We highlight our main contributions below:

- We design a family of $\ell_r$-norm ($r \in (1, 2)$) regularizers for OMD that allows interpolation between the negative entropy and squared Euclidean norm, adjusting its geometry to the sparsity of the cost functions (see Section 4). The regularizer naturally plugs into the standard OMD analysis.

- We show OMD with the above regularizer achieves sparsity-adaptive regret bounds of order $\mathcal{O}\left(\sqrt{DKT_\star \log MT_\star}\right)$ (Theorem 4.1). We also give a parameter-free version achieving the same bound (Theorem 4.4) that does not require prior knowledge of the sparsity level $M$ nor the expected hitting time of the optimal policy $T_\star$ (the only unknown parameters).

- We establish a lower bound of order $\Omega\left(\sqrt{DKT_\star \log M}\right)$ (Theorem 4.6), matching our regret guarantees up to a logarithmic factor of $T_\star$ (already present in prior work [8]) and improving over [8] in the $M = SA$ non-sparse setting by including the $\sqrt{\log SA}$ dependence.

---

[2]The experts setting is the single-state full-information feedback $A$-action online learning problem [5, 14].

- We show that OMD with the negative entropy used in prior work [28, 8] suffers regret at least $\Omega(\sqrt{K \log S})$ even when $M = 3$ (Theorem 3.1). This rules out the negative entropy as a viable regularizer in the sparse setting and provides justification for the use of our regularizer.

- We establish that results independent of $SA$ are not achievable in the unknown transitions setting via a lower bound in the sparse ($M = 1$) setting of order $\Omega\left(D\sqrt{SAK}\right)$ (Theorem 5.1).

**Technical Contributions:** Proving these results requires new technical ideas. For the general sparse lower-bound, we derive a result on the expectation of the maximum of asymmetric zero-mean random walks, generalizing the result for the symmetric case from [25]. The negative-entropy-specific lower-bound relies on the careful design of an MDP with skewed initial occupancy measures that highlights both the reasons for the failure of the negative entropy as well as the more general difficulty of the stochastic nature of SSP problems.

## 1.2 Related works

**Regret minimisation for SSP problems under full-information feedback** was initiated by a line of work studying stochastic costs [32, 29, 33, 11, 7, 9, 17]. In the adversarial setting, it was first studied by [28] in the known transition case. Their bounds were later improved by [8]. There have since been many extensions: [6] consider the unknown transition setting, [38] establish dynamic regret bounds, [10] consider a policy optimisation approach in the unknown transition setting.

**Regret minimisation for SSP problems under bandit feedback** where only the costs of the visited state-action pairs in an episode are revealed to the learner has also been studied both in the stochastic [10] and adversarial settings [8, 6, 10]. In the adversarial known transition setting, the minimax regret is of the order $\sqrt{KDT_\star SA}$ (ignoring log terms) [8]. It is an interesting future direction to study the sparse SSP problem with bandit feedback and understand if the regret scales with $\sqrt{M}$ instead of $\sqrt{SA}$, in which case $M$ would play the same role of effective dimension as in the setting we consider.

**Regret minimisation with sparse costs** was studied in the classical online learning setting [19] ($S = 1$). The minimax regret goes from $\mathcal{O}(\sqrt{K \log A})$ to $\mathcal{O}(\sqrt{KMA^{-1} \log A})$ under full-information feedback (experts problem). For rewards instead of costs, it goes from $\mathcal{O}(\sqrt{K \log A})$ to $\mathcal{O}(\sqrt{K \log M})$, which matches the benefits of sparsity we establish for the SSP problem with costs. Note that we restrict our focus to costs since it is unclear how to interpret rewards within the SSP framework. Under bandit feedback, the sparse minimax regret goes from $\widetilde{\mathcal{O}}(\sqrt{KA})$ to $\widetilde{\mathcal{O}}(\sqrt{KM})$ for both rewards and costs [19, 4]. The above minimax regrets can also be achieved by sparse-agnostic methods [19, 34]. Finally, sparsity was also considered in the case of stochastic losses by [20].

## 2 Preliminaries

### 2.1 Problem setting

We consider the *Stochastic Shortest Path* (SSP) problem with adversarial costs. The environment is modeled as a Markov Decision Process (MDP) $\mathcal{M} = (\mathcal{S}, \mathcal{A}, P, s_0, g)$ along with a sequence of cost functions $\{c_k\}_{k=1}^K$ chosen by an oblivious adversary over $K$ episodes. $\mathcal{S}$ is the state space with cardinality $S = |\mathcal{S}|$, and $s_0 \in \mathcal{S}$ is the fixed starting state. The goal state $g$ is a special absorbing state not included in $\mathcal{S}$. $\mathcal{A}$ is the action space with cardinality $A = |\mathcal{A}|$ and we assume for simplicity that it is the same in every state. Let $\Gamma = \mathcal{S} \times \mathcal{A}$ denote the set of all state-action pairs. The dynamics in the MDP are given by the *known* transition function $P$, where $P(s'|s, a)$ specifies the probability of moving to state $s' \in \mathcal{S} \cup \{g\}$ after taking action $a$ in state $s$.

Each episode begins in state $s_0$ and proceeds with the learner selecting actions until the goal state $g$ is reached. When the goal state is reached, the current episode ends and a new one begins. At the start of each episode $k$, the adversary selects a cost function $c_k : \Gamma \to [0, 1]$, assigning a cost to each state-action pair. We denote the sparsity level as $M = \max_k \sum_{(s,a) \in \Gamma} \mathbb{I}\{c_k(s,a) > 0\}$ the maximum number of non-zero costs in an episode. We work in the full-information setting where the entire function $c_k$ is revealed to the learner at the end of the episode. The objective is to minimize the total cost over all episodes, which requires a balance of minimizing the accumulated costs while ensuring the goal state is reached efficiently.

We use super-scripts to denote the time-step within an episode and sub-scripts to denote the episode: e.g. $(s_k^t, a_k^t)$ refers to the state-action pair at the $t$-th time-step of the $k$-th episode. We sometimes omit the sub-script when referring to an arbitrary episode. We now define some important concepts:

- A **stationary policy** $\pi$ is a mapping such that $\pi(\cdot|s)$ is a probability distribution over the choice of action $a \sim \pi(\cdot|s)$ in state $s$. A policy is called **proper** if it reaches the goal $g$ in finite time from any initial state in $\mathcal{S}$ with probability one, and improper if not. Let $\Pi_p$ be the set of all stationary proper policies. We assume the existence of at least one proper policy.

- The **expected hitting time** $T^\pi(s)$ is the expected number of steps required to reach $g$ from state $s$ under $\pi$. Letting $I_\pi(s)$ be the random number of time-steps used to reach the goal state when executing a policy $\pi$ in an episode starting from state $s$, then $T^\pi(s) = \mathbb{E}[I_\pi(s)]$. For any proper policy $\pi$, $I_\pi(s)$ and $T^\pi(s)$ are finite for all $s \in \mathcal{S}$.

- The **fast policy** $\pi_f$ is the deterministic policy that minimizes the worst-state expected hitting time, and the **diameter** $D$ of the MDP is the corresponding expected hitting time:

$$\pi_f = \arg\min_{\pi \in \Pi_p} \max_{s \in \mathcal{S}} T^\pi(s), \qquad D = \max_{s \in \mathcal{S}} T^{\pi_f}(s) = \max_{s \in \mathcal{S}} \min_{\pi \in \Pi_p} T^\pi(s).$$

Since the transition function $P$ is known, both the fast policy $\pi_f$ and the diameter $D$ can be computed offline prior to the learning process. We assume $D \geqslant 1$.

- The **cost-to-go** function $J_c^\pi : \mathcal{S} \to [0, \infty)$ is the expected cost suffered during an episode executing policy $\pi$ and starting from state $s$, given a cost function $c$ and a proper policy $\pi$. It is defined as

$$J_c^\pi(s) = \mathbb{E}\Big[ \sum_{t=1}^{I_\pi(s)} c(s^t, a^t) \Big| P, \pi, s^1 = s \Big],$$

where the expectation is with respect to the randomness in the action sampling and state transitions. We use $J_k^\pi$ to denote the cost-to-go from the initial state $s_0$ using the cost function $c_k$ in episode $k$.

- The **regret** $R_K$ is the primary measure of performance by which the learner is evaluated. It is the difference between the total cost over all episodes of the policies $\pi_1, \ldots, \pi_K$ chosen by the learner, and the total cost of the best proper deterministic policy in hindsight, $\pi^\star \in \arg\min_{\pi \in \Pi_p} \sum_{k=1}^K J_k^\pi$:

$$R_K = \sum_{k=1}^K \sum_{t=1}^{I_{\pi_k}(s_0)} c_k(s_k^t, a_k^t) - \sum_{k=1}^K J_k^{\pi^\star}.$$

- The **occupancy measure** $q_\pi \in \mathbb{R}_{\geqslant 0}^\Gamma$ of a proper policy $\pi$ is the expected number of visits to state-action pairs in an episode executing policy $\pi$ starting from $s_0$:

$$q_\pi(s, a) = \mathbb{E}\Big[ \sum_{i=1}^{I_\pi(s_0)} \mathbb{I}\big\{s^i = s, a^i = a\big\} \Big| P, \pi, s^1 = s_0 \Big].$$

The marginal $q_\pi(s) = \sum_{a \in \mathcal{A}} q_\pi(s, a)$ gives the expected number of visits to state $s$. Given a vector $q \in \mathbb{R}_{\geqslant 0}^\Gamma$, if it corresponds to a valid occupancy measure, the corresponding policy $\pi_q$ can be recovered via normalization as $\pi_q(a|s) = q(s, a)/\sum_{a'} q(s, a')$ [39, 28].

## 2.2 SSP as online linear optimisation and online mirror descent

Occupancy measures allow the cost-to-go to be expressed in a linear form:

$$J_k^\pi = \sum_{(s,a) \in \Gamma} q_\pi(s, a) c_k(s, a) = \langle q_\pi, c_k \rangle.$$

If the learner executes a stationary proper policy $\pi_k$ in episode $k$, the expected regret can thus be reformulated as an online linear optimisation problem on the space of occupancy measures:

$$\mathbb{E}\big[R_K\big] = \sum_{k=1}^K \Big\{ J_k^{\pi_k} - J_k^{\pi^\star} \Big\} = \sum_{k=1}^K \langle q_{\pi_k} - q_{\pi^\star}, c_k \rangle.$$

Online linear optimisation is a well studied problem and can be solved using Online Mirror Descent (OMD) (see e.g. [24]). In the SSP framework, OMD is applied on the space of occupancy measures corresponding to proper policies with expected hitting time bounded by some $T > 0$ defined as:

$$\Delta(T) = \Big\{ q \in \mathbb{R}^\Gamma : \sum_{(s,a) \in \Gamma} q(s, a) \leqslant T, \quad \forall s \in \mathcal{S} : \sum_{a \in \mathcal{A}} q(s, a) - \sum_{(s',a') \in \Gamma} P(s\,|\,s', a') q(s', a') = \mathbb{I}\{s = s_0\} \Big\}.$$

The first constraint ensures the expected hitting time is bounded by $T$, while the second is a flow constraint ensuring the vector corresponds to the occupancy measure of a policy. The regret bounds of OMD will hold against any fixed comparator policy as long as $T$ is large enough such that $\Delta(T)$ contains the occupancy measure of the optimal policy, i.e. $q_{\pi^\star} \in \Delta(T)$ or $T \geqslant T_\star$ where we denote by $T_\star = T^{\pi^\star}(s_0)$ the expected hitting time of $\pi^\star$. OMD with a strictly convex differentiable regularizer $\psi$ and step-size $\eta$ selects occupancy measures computed through the update

$$q_1 = \arg\min_{q \in \Delta(T)} \psi(q), \qquad q_{k+1} = \arg\min_{q \in \Delta(T)} \Big\{ \eta \cdot \langle q, c_k \rangle + D_\psi(q, q_k) \Big\}, \tag{1}$$

where $D_\psi(x, y) = \psi(x) - \psi(y) - \langle \nabla\psi(y), x - y \rangle$ is the Bregman divergence with respect to $\psi$. This update can be computed efficiently for all the regularizers we will discuss (see Appendix B). As discussed in the previous section, we can easily recover via normalization the corresponding policy $\pi_{q_k}$ that will be executed by the learner.

If the regularizer satisfies for some $\alpha > 0$, any $q \in \mathbb{R}^\Gamma$ and all $k \geqslant 1$:

$$\nabla\psi(q) \in [\nabla\psi(q_k), \nabla\psi(q_k) - \eta c_k] \implies \nabla^2\psi(q) \geq \alpha\nabla^2\psi(q_k), \tag{2}$$

(this is satisfied by many common regularizers), then a standard result (see e.g. Theorem 6 in [4], Theorem 5.5 in [3]) gives the following general regret bound for OMD:

$$\sum_{k=1}^{K} \langle q_k - q_{\pi^\star}, c_k \rangle \leqslant \underbrace{\frac{\psi(q_{\pi^\star}) - \psi(q_1)}{\eta}}_{\text{Penalty}} + \underbrace{\frac{\eta}{2\alpha} \sum_{k=1}^{K} \|c_k\|_{\nabla^2\psi(q_k)^{-1}}^2}_{\text{Stability}} \tag{3}$$

where $\|q\|_A^2 = \sum_{s,a,s',a'} q(s, a) A((s, a), (s', a')) q(s', a')$ for a matrix $A \in \mathbb{R}^{\Gamma \times \Gamma}$. Various regret bounds can be obtained by instantiating the above with different regularizers. In particular, [8] use the negative-entropy to obtain a $\mathcal{O}\left(\sqrt{DKT_\star \log SAT_\star}\right)$ regret bound.

## 3 Failure of negative entropy regularization

In the general non-sparse setting, [8] use the negative entropy to achieve a regret of $\mathcal{O}\left(\sqrt{DKT_\star \log SAT_\star}\right)$, which in the non-sparse setting has optimal dependence on $SA$ (as we show later in Theorem 4.4). Despite this success, the negative entropy fails to benefit from sparsity in its dependence on $S$, as shown by the result below. As we will see in Section 4, this establishes the negative entropy as a sub-optimal choice for sparse SSP problems.

**Theorem 3.1.** *For any $S \geqslant 6$, there exists an SSP instance with a fixed horizon of 3, sparsity level $M = 3$, an action space of size $A = 2$ and state space of size $S$ such that the regret of OMD (1) with negative-entropy regularization and any step-size $\eta > 0$ after $K$ episodes is $\mathbb{E}[R_K] = \Omega\left(\min\{\sqrt{K \log S}, K\}\right)$.*

This result shows that despite the SSP instance being sparse ($M = 3$), the regret of OMD with negative entropy regularization nevertheless scales as $\sqrt{\log S}$, which is the same dependence on $S$ as in the non-sparse setting. For sparse problems, the negative entropy provides no improvement on the regret with respect to $S$. This highlights that existing approaches and regularizers are inadequate to appropriately exploit sparse problems and motivates considering alternate regularizers specifically designed for the geometry of sparse problems, as we do in the next section.

To better understand the failure of negative entropy regularization in sparse settings, we highlight the main intuition behind the lower bound construction and defer the details of the proof to Appendix A.

**Proof intuition:** The key idea is to reduce SSP to an experts problem with 2 actions and a heavily skewed initial distribution over the actions. The initial occupancy measure played by OMD in (1) is $q_1 = \arg\min_{q \in \Delta(T)} \psi(q)$. For most regularizers, including the negative entropy, this encourages $q_1$ to be uniform across the state-action space while maintaining the constraints on the flow and expected hitting time. Since we consider a fixed-horizon MDP, only the flow constraint is relevant.

Consider the SSP problem shown in Figure 1 with $N = S - 2$. Since $s_1, ..., s_N$ constitute a large majority of the states (especially for large $N$), $\psi(q)$ is mainly affected by the values of $q$ in these

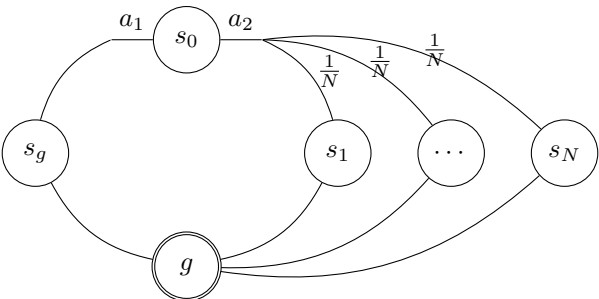

Figure 1: MDP for the reduction to a skewed experts problem with 2 actions: $\mathcal{S} = \{s_0, s_g, s_1, ..., s_N\}$ $(N = S - 2)$, $\mathcal{A} = \{a_1, a_2\}$. The transitions are given by $p(s_g|s_0, a_1) = 1, p(g|s_g, a) = 1$ for all $a \in \mathcal{A}$, for $i \geqslant 1$: $p(s_i|s_0, a_2) = 1/N, p(g|s_i, a) = 1$ for all $a \in \mathcal{A}$.

states. In order to minimize $\psi(q)$, $q_1$ needs to ensure the expected number of visits to these $N$ states is sufficiently high. However since for any $q$ and any $i \geqslant 1$, $q(s_i) = \frac{1}{N}q(s_0, a_2)$, for $q_1(s_i)$ to be sufficiently large then $q_1(s_0, a_2)$ needs to be much larger (by a factor of $N$). This results in $q_1$ being heavily skewed towards $a_2$ in $s_0$. For the negative entropy, this gives specifically $q_1(s_0, a_1) \approx \frac{1}{\sqrt{N}}$.

If the costs in all states but $s_0$ are set to 0, the problem is sparse ($M = 2$) and reduces to a experts problem with 2 actions where the initial probability for the first action, which in our case is $q_1(s_0, a_1)$, scales as $1/\sqrt{N}$. The regret for OMD with the negative entropy in this setting can be shown to scale for any step-size at least as $\Omega(\sqrt{K \log N}) = \Omega(\sqrt{K \log S})$, providing the dependence from the statement of the theorem. To prove this formally for the SSP reduction, we use the above construction coupled with a non-skewed reduction and careful setting of the costs. We include the details in Appendix A.

Finally, we remark that this failure comes from the negative entropy stretching euclidean distance near the boundary of the space in such a way that two nearby points in terms of euclidean distance can be arbitrarily far in terms of negative entropy. This makes it hard for OMD to recover from the initial occupancy measure $q_1(s_0, a_1) \approx 1/\sqrt{N}$ ($\rightarrow 0$ as $N$ increases) unless the step-size is unreasonably large. This property does not generalize to all regularizers and in fact provides insights for designing a regularizer to appropriately handle sparsity. In particular, the regularizer we consider in the next section does not suffer from the same issue because the stretching of euclidean distance is finite since its gradient does not diverge at the boundary (i.e. as $q(s, a) \rightarrow 0$) unlike the negative entropy.

## 4 The benefits of sparsity

In this section, we show that it is possible to achieve a regret bound of order $\mathcal{O}\big(\sqrt{DKT_\star \log(MT_\star)}\big)$, where $M$ is the maximum number of non-zero entries in the cost. This is our main result and together with the lower bound in Theorem 4.6 establishes that the sparsity level $M$ acts as a measure of effective dimension instead of the state-action space size $SA$ for SSP with full-information feedback.

In the previous section, we showed and discussed that the negative entropy, the regularizer used in OMD by existing methods, is inadequate to handle sparse SSP problems. Motivated by this failure, we consider alternate regularizers. However, identifying a suitable regularizer poses two key challenges. Firstly, it must work for SSP and the associated technical complexities compared to other simpler online learning problems. In particular, it needs to match the dependence in terms of the other non-sparsity-related quantities appearing in the regret of the negative entropy (i.e. $D, T_\star, K$). Second, it must explicitly leverage sparsity to improve performance. We propose the following family of regularizers parameterised by $p > 1$:

$$\psi_p(q) = p \cdot \left(-1 + \|q\|_{1+1/p}^{1+1/p}\right) = p \cdot \left(-1 + \sum_{s \in \mathcal{S}} \sum_{a \in \mathcal{A}} |q(s, a)|^{1+1/p}\right). \tag{4}$$

As $p \rightarrow \infty$, the regularizer in Equation (4) converges to the negative entropy. On the other hand, as $p \rightarrow 1$, the regularizer converges to the squared Euclidean norm that enforces much weaker regularization on sparse points. Therefore, $\psi_p$ allows smooth interpolation between dense and sparse regimes via the tunable parameter $p$ (see Figure 2 for a comparison). In particular, $\psi_p$ for small $p$

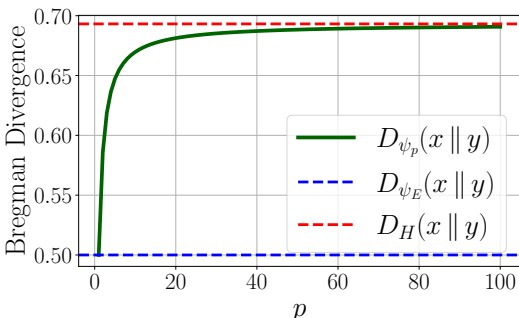

Figure 2: Bregman Divergence between a deterministic distribution $x = [0, 1]$ and the uniform distribution $y = [1/2, 1/2]$ for our regularizer $\psi_p$, squared Euclidean norm $\psi_E$ and negative entropy $H$ for increasing values of $p$.

induces an OMD update that is able to easily move to and away from the boundary of the space, making it robust to the skewed initial occupancy measure on the SSP instance which caused the failure of the negative entropy in Section 3. The parameter $p$ also controls a trade-off between the stability and penalty term in (3), which ultimately will enable the removal of the dependence on $SA$.

Versions of this family of regularizers can be found in the convex optimization literature [18, 22]. As far as we are aware, its use with OMD is novel. A regularizer involving an $r$-norm with $r \in (1, 2]$ has been used but the norm is squared rather than to the power of $r$ (see e.g. Section 6.7 in [24]). Our regularizer is also similar in flavor to the Tsallis-entropy in the sense that it converges to the negative entropy in the limit of its parameter.

We note that OMD with the above regularizer can be implemented efficiently for any $p$: the projection step of OMD over $\Delta(T)$ can be written as a convex optimization problem as in [28], which can be solved efficiently (details in Appendix B).

We can now turn to our main result, which establishes regret bounds that scale with the sparsity level $M$ for OMD with the regularizer in (4) when $M$ is known.

**Theorem 4.1.** *Consider OMD with $\psi_p$ as regularizer. If $T > e$ is such that $q_{\pi^\star} \in \Delta(T)$, $\eta = \sqrt{\frac{pT^{1+1/p}}{KDM^{1/p}}}$, $p = \log(TM)$, then $\mathbb{E}[R_K] \leqslant \mathcal{O}\left(\sqrt{DKT \log(MT)}\right)$.*

We present below the outline of the proof and include the missing details in Appendix C.

*Proof.* It can be shown that $\psi_p$ satisfies the condition (2) with $\alpha = 1$, allowing us to use the bound in (3) as a starting point. Using that $q_\star \in \Delta(T)$, we can bound the Penalty term:

$$\psi_p(q_\star) - \psi_p(q_1) = p\left(\|q_\star\|_{1+1/p}^{1+1/p} - \|q_1\|_{1+1/p}^{1+1/p}\right) \leqslant p \cdot \|q_\star\|_1^{1+1/p} \leqslant p \cdot T^{1+1/p} \tag{5}$$

It can also be shown that $\nabla^2 \psi_p(q)^{-1} = \text{diag}\left(\frac{p}{p+1} q^{1-1/p}\right)$. Using that $c_k(s, a)^2 \leqslant c_k(s, a)$, we get

$$\|c_k\|_{\nabla^2 \psi_p(q_k)^{-1}}^2 \leqslant \frac{p}{p+1} \sum_{s,a} c_k(s, a) q_k(s, a)^{1-1/p} \leqslant \|c_k\|_1 \sum_{s,a} \frac{c_k(s, a)}{\|c_k\|_1} q_k(s, a)^{1-1/p}$$

$$\leqslant \|c_k\|_1 \left(\sum_{s,a} \frac{c_k(s, a)}{\|c_k\|_1} q_k(s, a)\right)^{1-1/p} \tag{6}$$

$$= \|c_k\|_1^{1/p} \langle c_k, q_k \rangle^{1-1/p} \leqslant M^{1/p} \max\{1, \langle c_k, q_k \rangle\} \leqslant M^{1/p}\left(1 + \langle c_k, q_k \rangle\right),$$

where the key step (6) uses Jensen's inequality on the concave function $x^{1-1/p}$ $(p > 1)$ and probability distribution $c_k/\|c_k\|_1$. Plugging this into the Stability term and combining with (5):

$$\sum_{k=1}^K \langle q_k - q_{\pi^\star}, c_k \rangle \leqslant \frac{pT^{1+1/p}}{\eta} + \frac{\eta M^{1/p} K}{2} + \frac{\eta M^{1/p}}{2} \sum_{k=1}^K \langle q_k, c_k \rangle$$

$$\implies \sum_{k=1}^{K} \langle q_k - q_{\pi^\star}, c_k \rangle \leqslant \frac{1}{1 - \frac{\eta M^{1/p}}{2}} \left( \frac{pT^{1+1/p}}{\eta} + \frac{\eta M^{1/p} K}{2} + \frac{\eta M^{1/p}}{2} \sum_{k=1}^{K} \langle q_\star, c_k \rangle \right)$$

$$\implies \sum_{k=1}^{K} \langle q_k - q_{\pi^\star}, c_k \rangle \leqslant \frac{2pT^{1+1/p}}{\eta} + 2\eta M^{1/p} DK,$$

where the last step uses that $\sum_{k=1}^{K} \langle q_\star, c_k \rangle \leqslant \sum_{k=1}^{K} \langle q_{\pi_f}, c_k \rangle \leqslant K \|q_{\pi_f}\|_1 \|c_k\|_\infty \leqslant DK$, $D \geqslant 1$ and $\eta \leqslant 4M^{-1/p} \iff 1 - \eta M^{1/p}/2 \geqslant 1/2$. Tuning $\eta = \sqrt{\frac{pT^{1+1/p}}{KDM^{1/p}}}$ (so $\eta \leqslant 4M^{-1/p}$ for sufficiently large $K$) and $p = \log(TM)$ gives the result. $\square$

Provided we can suitably select $T \approx T_\star$ (see Section 4.1), this result establishes that sparsity does lead to an improvement in the minimax regret. In Section 4.2, we show that the dependence on $M$ and $SA$ is optimal, ruling out polynomial improvements from sparsity such as in the experts setting [19]. This highlights that $M$ acts as the effective dimension of the problem instead of $SA$. In particular, if the sparsity level $M$ is constant, then we obtain a dimension-independent regret of $\mathcal{O}(\sqrt{DKT \log T})$.

**Remark 4.2.** *Although we express the bound in terms of the sparsity level $M$, it can be seen that the analysis above holds more generally if $M$ is instead an upper bound on the $\ell_1$ norm of the costs: $M = \max_k \|c_k\|_1$. This relaxation allows our result to cover "softly sparse" cost structures and aligns with the notion of first-order bounds commonly studied in the online learning literature [23, 37, 35].*
**Remark 4.3.** *The above result does not recover the $M/A$ polynomial improvement in the special case of the expert setting. This can be recovered through a regret bound of a slightly different flavor which includes the hitting time of the uniform policy. We include the details and subtleties in Appendix D but the upshot is that the necessity to reach the goal state in SSP creates a fundamental difference in the benefits of sparsity compared to the expert setting.*

### 4.1 Sparse-agnostic parameter-free upper bound

The procedure in Theorem 4.4 assumes knowledge of the sparsity level $M$ to tune the parameter $p$ of our regularizer and uses knowledge of the expected hitting time of the optimal policy $T_\star$ to consider OMD over the space of suitable occupancy measures. We now adapt existing techniques to remove both of these assumptions and derive fully parameter-free guarantees.

For the unknown sparsity level, we use the same approach as in [19]. We divide the $K$ episodes into batches. Within each batch, we independently run OMD tuning the parameter $p$ of our regularizer with the sparsity level observed up to the current batch, as described in Algorithm 1 in Appendix C.2.

For the unknown expected hitting time of the optimal policy $T_\star$, we can exploit the same meta-algorithm technique as in [8], using the sparse agnostic algorithms introduced above as base learners. We run $N \approx \log K$ instances of Algorithm 1 where the $j$-th instance sets its parameter $T$ as $b(j) \approx 2^j$. Therefore, there exists a good instance $j_\star$ such that $b(j_\star)$ is close to the unknown $T_\star$. The regret of a scale-invariant meta-algorithm, described for completeness in Algorithm 2 in Appendix C.2, closely matches that of this good instance.

Together, these two techniques yield the following parameter-free regret bound (proof in Appendix C.2):

**Theorem 4.4.** *If $K > \max\left(T_\star, \frac{T_\star}{D} \log(T_\star M)\right)$ and $T_\star > e$, Algorithm 2 guarantees $\mathbb{E}\left[R_K\right] \leqslant \tilde{\mathcal{O}}\left(\sqrt{DKT_\star \log(MT_\star)} + T_\star\right)$, where the notation $\tilde{\mathcal{O}}$ hides double-logarithmic factors.*

The leading term matches the regret bound from Theorem 4.1, while the second does not depend on $M$ or $K$. Therefore, running a procedure that does not assume knowledge of $M$ and $T_\star$ comes at no additional cost in terms of the regret bound (up to double-logarithmic factors). We also note that it is common for log-log factors to be ignored in parameter free results with expert-like algorithms [12, 16]. It is also possible to obtain a bound that holds with high-probability since the high-probability analysis given in [8] can easily be adapted to work with our regularizer.

**Remark 4.5.** *The assumption $K \geqslant \frac{T_\star}{D} \log(T_\star M)$ or $K \geqslant \frac{T_\star}{D} \log(T_\star SA)$ in the non-sparse setting is actually non-restrictive since it is required for the upper-bound to be meaningful:*

$$\sqrt{DKT_\star \log(MT_\star)} \leqslant T_\star K \iff K \geqslant \frac{T_\star}{D} \log(T_\star M).$$

*In particular, it is likely that there is a gap between the behavior of the minimax regret between the "low-dimensional" setting which we study and a high-dimensional setting where $K \ll \frac{T_\star}{D} \log(T_\star M)$. The high-dimensional problem is yet to be explored, even in the non-adversarial setting and could be an interesting avenue of future research. Indeed, all prior works on SSP have implicitly studied the problem in low-dimension, which comes with an implicit assumption that $K$ is sufficiently large.*

### 4.2 Lower bound

In this section, we provide a general lower bound for sparse SSP problems.

**Theorem 4.6.** *For any $D, T_\star, K, S, A$ with $T^\star \geqslant D \geqslant 3 \log S$, $S(A-1) \geqslant 400$, $K \geqslant \frac{800 T^\star}{D} \log M$ and $M \geqslant 101$, there exists an SSP instance with stochastic $M$-sparse costs, $S$ states and $A$ actions such that its diameter is $D$, the expected hitting time of the optimal policy is $T^\star$, and the expected regret with respect to the randomness of the losses for any learner after $K$ episodes is $\mathbb{E}[R_K] \geqslant \Omega\big(\sqrt{KT^\star D \log M}\big)$.*

For general $M$ ($> 100$), the lower bound matches the upper bound established in Theorem 4.1 in its dependence on $M$, characterizing the minimax regret for general sparse problems (up to a $\log T_\star$ term). For $M = SA$, our result gives a $\Omega\big(\sqrt{KT^\star D \log SA}\big)$ lower bound improving on the $\Omega\big(\sqrt{KT^\star D}\big)$ lower bound of [8]. In particular, this establishes the optimal dependence on the size of the state-action space $SA$ in the minimax regret for the general non-sparse SSP problem.

**Proof intuition:** The proof is based on the combination of an SSP instance from [8] and a probabilistic costs construction, which then requires some non-trivial arguments to extend to the sparse SSP problem. We give an overview of the construction and defer the details to Appendix E.

The MDP construction is essentially a reduction to a non-sparse experts problem with $\mathcal{O}(M)$ actions. First, there is a reduction to an experts problem with $\mathcal{O}(SA)$ actions. Then within these, there are $\mathcal{O}(M)$ good actions, while the remaining are bad. The good actions suffer small costs in expectation and can lead directly to the goal-state. The bad actions are zero-cost but all lead to the same unique bad state, where only one action leads to the goal-state and suffers high cost. This allows a big proportion of the actions to be bad while still guaranteeing sparsity and forcing the learner to only consider the $\mathcal{O}(M)$ good actions, completing the reduction to the non-sparse experts problem with $\mathcal{O}(M)$ actions.

However, we cannot directly apply lower bounds for the experts problem because of subtleties in the reduction and the cost-generating mechanism. We use a similar approach to the experts lower bounds by sampling the costs i.i.d. from a Bernoulli distribution, however with a scaled parameter to ensure the reduction above holds. The regret in this stochastic environment can then be expressed as the maximum of asymmetric zero-mean i.i.d. random walks, capturing how much better the optimal policy can be by choosing the best action after the i.i.d. Bernoulli costs have been sampled for all episodes. The result then follows from a technical result on the expectation of this maximum that we derive in Appendix G. We note that the reduction and costs are constructed in such a way that the diameter of the MDP and expected hitting time of the optimal policy are indeed $D$ and $T_\star$.

## 5 Unknown transition setting

In this section, we consider the setting where the transitions are unknown and show through the following lower bound that the benefits of sparsity are limited.

**Theorem 5.1.** *For any $D, K, S, A$ with $S \geqslant 2, A \geqslant 16$, $D \geqslant 2$ and $K \geqslant SA$, there exists an SSP instance with $M = 1$, $S$ states and $A$ actions such that its diameter is $D$ and the expected regret for any learner without knowledge of the transitions after $K$ episodes is $\mathbb{E}[R_K] \geqslant \Omega\big(D\sqrt{SAK}\big)$.*

The above result establishes that the minimax regret for the unknown transition setting must scale polynomially with $SA$, regardless of the sparsity. In particular, this highlights the limited benefits of sparsity in removing the dependence on the state-action space size in the unknown transition setting, which is in stark contrast to the known transition setting.

The proof is based on an SSP instance used by [29] to prove an $\Omega\big(D\sqrt{SAK}\big)$ lower bound in the unknown transition non-sparse setting. It turns out that this instance can be adapted such that the cost is only non-zero for a single state-action pair, while keeping the regret lower bound unchanged, giving the above result. We include the details in Appendix F.

# 6   Conclusion, limitations and future-work

In this work, we studied the SSP problem under sparse adversarial costs and full-information feedback. When the transitions are known, we have shown that existing methods fail to appropriately exploit sparsity. Instead, we designed a family of regularizers to use with Online Mirror Descent that allowed us to characterize the sparse minimax regret, establishing the extent of the benefits of sparsity in this setting. When the transitions are unknown, we showed that even the sparse minimax regret scales polynomially in the size of the state-action space, suggesting fundamental limits in such settings.

Our results open up many further directions of research. In particular, we established the benefits of sparsity under known transition as limited to logarithmic, however, there could be structural properties of an MDP that could break this logarithmic limit and achieve polynomial benefits. Moreover, we have limited our focus to the adversarial full-information feedback setting, but the study of sparse SSP problems in other settings, such as partial feedback, stochastic environments, or structured decision problems remains unexplored.

## Acknowledgments and Disclosure of Funding

Emmeran Johnson is funded by EPSRC through the Modern Statistics and Statistical Machine Learning (StatML) CDT (grant no. EP/S023151/1). Alberto Rumi was funded by European Lighthouse of AI for Sustainability project (ELIAS). Patrick Rebeschini was funded by UK Research and Innovation (UKRI) under the UK government's Horizon Europe funding guarantee (grant no. EP/Y028333/1).

We would like to thank the reviewers and meta-reviewers for their time and feedback.

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

## A  Failure of the negative entropy

In this appendix, we prove our lower bound result for OMD with the negative entropy from Section 3. We first restate the result.

**Theorem 3.1.** *For any $S \geqslant 6$, there exists an SSP instance with a fixed horizon of $3$, sparsity level $M = 3$, an action space of size $A = 2$ and state space of size $S$ such that the regret of OMD (1) with negative-entropy regularization and any step-size $\eta > 0$ after $K$ episodes is $\mathbb{E}[R_K] = \Omega(\min\{\sqrt{K \log S}, K\})$.*

*Proof.* Fix $K$ even, $S \geqslant 6$, $A = 2$ and $N = S - 5$. We first describe the SSP instance. Consider the following MDP $\mathcal{M} = (\mathcal{S}, \mathcal{A}, p, s_0, g)$, where $\mathcal{S} = \{s_0, s_0^L, s_0^R, s_1^R, ..., s_N^R, s_1\}$ and $\mathcal{A} = \{a_1, a_2\}$. The transitions and costs (in each episode $k$) are defined as:

- $s_0$: $p(s_0^L|s_0, a) = p(s_0^R|s_0, a) = 1/2$ and $c_k(s_0, a) = 0$ for all $a \in \mathcal{A}$.

- $s_0^L$: $p(s_g^L|s_0^L, a) = 1$ for all $a \in \mathcal{A}$ and $c_k(s_0^L, a_1) = \frac{1+(-1)^k}{2}$, $c_k(s_0^L, a_2) = 1/2$.

- $s_0^R$: $p(s_g^R|s_0^R, a_1) = 1$, $p(s_i^R|s_0^R, a_2) = 1/N$ and $c_k(s_0^R, a_1) = 0$, $c_k(s_0^R, a_2) = 1$ .

- $s_i^R$: $p(g|s_i^R, a) = 1$ and $c_k(s_i^R, a) = 0$ for all $a \in \mathcal{A}$.

- $s_g^L$: $p(g|s_g^L, a) = 1$ and $c_k(s_g^L, a) = 0$ for all $a \in \mathcal{A}$.

- $s_g^R$: $p(g|s_g^R, a) = 1$ and $c_k(s_g^R, a) = 0$ for all $a \in \mathcal{A}$.

An illustration is given in Figure 3. This SSP instance has a fixed horizon of 3 in the sense that all policies have a hitting time of exactly 3 (the states $s_g^L$ and $s_g^R$ are added to guarantee this). As a result we have that $T_\star = D = 3$. Also note that there are at most 3 state-action pairs that have non-zero cost, therefore the sparsity level $M = 3$.

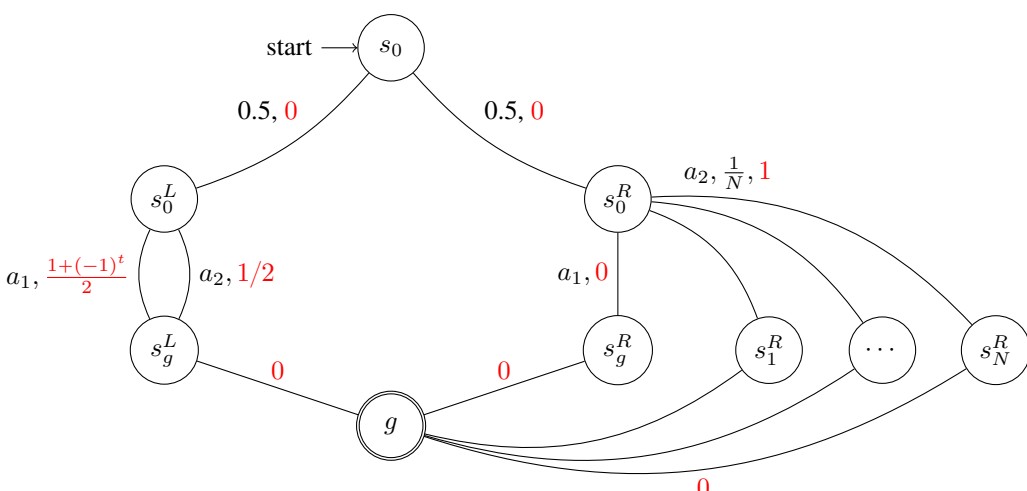

Figure 3: Diagram illustrating MDP construction for the proof of Theorem 3.1. When an action is not specified for an edge, then both actions give the same transition and cost. If an edge has a number in black, it is a transition probability; if it does not then the transition is deterministic. The costs are given in red. The formal description of the MDP is given above.

From Appendix B.1 in [28] (we can ignore the optimization over $\lambda$ because we are in a fixed horizon setting), the update of OMD with negative entropy for any $k \geqslant 0$ can be computed by solving a convex optimization problem:

$$q_{k+1}(s, a) = q_k(s, a)e^{B_k^{v_{k+1}}(s,a)}, \quad \text{where} \quad B_k^v(s, a) = v(s) - \eta c_k(s, a) - \sum_{s' \in \mathcal{S}} p(s'|s, a)v(s'),$$

$$v_{k+1} = \arg\min_v \mathcal{D}_k(v),$$

$$\mathcal{D}_k(v) = \sum_{s \in \mathcal{S}} \sum_{a \in \mathcal{A}} q_k(s, a)e^{B_k^v(s,a)} - v(s_0),$$

with $q_0(s, a) = 1$ and $c_0(s, a) = 0$. This allows us to compute exactly the points played by the algorithm on the SSP instance described above, and in turn compute the regret, from which the result will follow.

**In the following few pages, we compute the occupancy measures played by OMD with the negative entropy on the SSP instance described earlier for all episodes, using the convex optimization problem above.** We begin by computing expressions for $B_k^v(s, a)$ in each state:

- $B_k^v(s_0, a) = v(s_0) - \frac{1}{2}v(s_0^L) - \frac{1}{2}v(s_0^R)$ for all $a \in \mathcal{A}$

- $B_k^v(s_0^L, a) = v(s_0^L) - \eta c_k(s_0^L, a) - v(s_g^L)$ for all $a \in \mathcal{A}$

- $B_k^v(s_g^L) = v(s_g^L)$

- $B_k^v(s_0^R, a_1) = v(s_0^R) - v(s_g^R)$

- $B_k^v(s_0^R, a_2) = v(s_0^R) - \eta c_k(s_0^R, a_2) - \frac{1}{N}\sum_{i=1}^{N} v(s_i^R) = v(s_0^R) - \eta c_k(s_0^R, a_2) - v(s_1^R)$
  since by symmetry $v(s_i^R) = v(s_1^R)$ for all $i \geqslant 1$ and any $v$ solving the convex optimization problem specified in the OMD update.

- $B_k^v(s_i^R, a) = v(s_i^R) = v(s_1^R)$ for all $a \in \mathcal{A}$

- $B_k^v(s_g^R, a) = v(s_g^R)$ for all $a \in \mathcal{A}$

Plugging these into the optimization problem, we obtain (recall the notation $q(s) = \sum_{a \in \mathcal{A}} q(s, a)$):

$$v_{k+1} = \arg\min_v \quad \mathcal{D}_k(v) = \arg\min_v \sum_{s \in \mathcal{S}} \sum_{a \in \mathcal{A}} q_k(s, a)e^{B_k^v(s,a)} - v(s_0)$$

$$= \arg\min_v \quad q_k(s_0, a_1)e^{v(s_0)-0.5v(s_0^L)-0.5v(s_0^R)} + q_k(s_0, a_2)e^{v(s_0)-0.5v(s_0^L)-0.5v(s_0^R)}$$

$$+ q_k(s_0^L, a_1)e^{v(s_0^L)-\eta c_k(s_0^L,a_1)-v(s_g^L)} + q_k(s_0^L, a_2)e^{v(s_0^L)-\eta c_k(s_0^L,a_2)-v(s_g^L)}$$

$$+ q_k(s_g^L, a_1)e^{v(s_g^L)} + q_k(s_g^L, a_2)e^{v(s_g^L)}$$

$$+ q_k(s_0^R, a_1)e^{v(s_0^R)-v(s_g^R)} + q_k(s_0^R, a_2)e^{v(s_0^R)-\eta c_k(s_0^R,a_2)-v(s_1^R)}$$

$$+ \sum_{i=1}^{N}\left\{q_k(s_i^R, a_1)e^{v(s_1^R)} + q_k(s_i^R, a_2)e^{v(s_1^R)}\right\}$$

$$+ q_k(s_g^R, a_1)e^{v(s_g^R)} + q_k(s_g^R, a_2)e^{v(s_g^R)}$$

$$- v(s_0)$$

$$= \arg\min_v \quad q_k(s_0)e^{v(s_0)-0.5v(s_0^L)-0.5v(s_0^R)}$$

$$+ q_k(s_0^L, a_1)e^{v(s_0^L)-\eta c_k(s_0^L,a_1)-v(s_g^L)} + q_k(s_0^L, a_2)e^{v(s_0^L)-\eta c_k(s_0^L,a_2)-v(s_g^L)}$$

$$+ q_k(s_g^L)e^{v(s_g^L)}$$

$$+ q_k(s_0^R, a_1)e^{v(s_0^R)-v(s_g^R)} + q_k(s_0^R, a_2)e^{v(s_0^R)-\eta c_k(s_0^R,a_2)-v(s_1^R)}$$

$$+ \sum_{i=1}^{N}\left\{q_k(s_i^R)e^{v(s_1^R)}\right\}$$

$$+ q_k(s_g^R)e^{v(s_g^R)}$$

$$- v(s_0).$$

This being a convex optimization problem, it can be solved by differentiating and setting to 0:

$$\frac{\partial \mathcal{D}_k(v)}{\partial v(s_0)} = q_k(s_0)e^{v(s_0)-0.5v(s_0^L)-0.5v(s_0^R)} - 1 = 0$$

$$\frac{\partial \mathcal{D}_k(v)}{\partial v(s_0^L)} = -0.5q_k(s_0)e^{v(s_0)-0.5v(s_0^L)-0.5v(s_0^R)} + q_k(s_0^L, a_1)e^{v(s_0^L)-\eta c_k(s_0^L,a_1)-v(s_g^L)} + q_k(s_0^L, a_2)e^{v(s_0^L)-\eta c_k(s_0^L,a_2)-v(s_g^L)}$$

$$= -0.5 + q_k(s_0^L, a_1)e^{v(s_0^L)-\eta c_k(s_0^L,a_1)-v(s_g^L)} + q_k(s_0^L, a_2)e^{v(s_0^L)-\eta c_k(s_0^L,a_2)-v(s_g^L)} = 0$$

$$\frac{\partial \mathcal{D}_k(v)}{\partial v(s_g^L)} = -q_k(s_0^L, a_1)e^{v(s_0^L)-\eta c_k(s_0^L,a_1)-v(s_g^L)} - q_k(s_0^L, a_2)e^{v(s_0^L)-\eta c_k(s_0^L,a_2)-v(s_g^L)} + q_k(s_g^L)e^{v(s_g^L)}$$

$$= q_k(s_g^L)e^{v(s_g^L)} - 0.5 = 0$$

$$\frac{\partial \mathcal{D}_k(v)}{\partial v(s_0^R)} = -0.5q_k(s_0)e^{v(s_0)-0.5v(s_0^L)-0.5v(s_0^R)} + q_k(s_0^R, a_1)e^{v(s_0^R)-v(s_g^R)} + q_k(s_0^R, a_2)e^{v(s_0^R)-\eta c_k(s_0^R,a_2)-v(s_1^R)}$$

$$= -0.5 + q_k(s_0^R, a_1)e^{v(s_0^R)-v(s_g^R)} + q_k(s_0^R, a_2)e^{v(s_0^R)-\eta c_k(s_0^R,a_2)-v(s_1^R)} = 0$$

$$\frac{\partial \mathcal{D}_k(v)}{\partial v(s_1^R)} = -q_k(s_0^R, a_2)e^{v(s_0^R)-\eta c_k(s_0^R,a_2)-v(s_1^R)} + e^{v(s_1^R)}\sum_{i=1}^{N} q_k(s_i^R) = 0$$

$$\frac{\partial \mathcal{D}_k(v)}{\partial v(s_g^R)} = -q_k(s_0^R, a_1)e^{v(s_0^R)-v(s_g^R)} + q_k(s_g^R)e^{v(s_g^R)} = 0.$$

**Let's look specifically at the case** $k = 0$ $(q_0(s, a) = 1, q_0(s) = 2, c_0(s, a) = 0$ for all $s \in \mathcal{S}, a \in \mathcal{A})$.
For the left part of the MDP we have:

$$\frac{\partial \mathcal{D}_k(v)}{\partial v(s_0^L)} = 0 \implies 2e^{v(s_0^L)-v(s_g^L)} = 0.5 \implies e^{v(s_0^L)} = 0.25e^{v(s_g^L)}$$

$$\frac{\partial \mathcal{D}_k(v)}{\partial v(s_g^L)} = 0 \implies e^{v(s_g^L)} = 0.25 \implies e^{v(s_0^L)} = 0.25^2$$

$$\implies q_1(s_0^L, a) = e^{B_0^v(s_0^L,a)} = \frac{0.25^2}{0.25} = 0.25, \text{ for all } a \in \mathcal{A}.$$

For the right part of the MDP, we have:

$$\frac{\partial \mathcal{D}_k(v)}{\partial v(s_0^R)} = 0 \implies e^{v(s_0^R)-v(s_g^R)} + e^{v(s_0^R)-v(s_1^R)} = 0.5 \implies e^{v(s_0^R)} = \frac{0.5}{e^{-v(s_g^R)} + e^{-v(s_1^R)}}$$

$$\frac{\partial \mathcal{D}_k(v)}{\partial v(s_1^R)} = 0 \implies e^{v(s_0^R)-v(s_1^R)} = 2Ne^{v(s_1^R)} \implies e^{v(s_1^R)} = \frac{1}{\sqrt{2N}}e^{0.5v(s_0^R)}$$

$$\frac{\partial \mathcal{D}_k(v)}{\partial v(s_g^R)} = 0 \implies e^{v(s_0^R)-v(s_g^R)} = 2e^{v(s_g^R)} \implies e^{v(s_g^R)} = \frac{1}{\sqrt{2}}e^{0.5v(s_0^R)}$$

$$\implies e^{v(s_0^R)} = \frac{0.5}{\sqrt{2}e^{-0.5v(s_0^R)} + \sqrt{2N}e^{-0.5v(s_0^R)}}$$

$$\implies e^{0.5v(s_0^R)} = \frac{0.5}{\sqrt{2} + \sqrt{2N}}$$

$$\implies e^{v(s_0^R)} = \frac{0.25}{(\sqrt{2} + \sqrt{2N})^2}$$

$$\implies q_1(s_0^R, a_1) = e^{B_0^v(s_0^R,a_1)} = e^{v(s_0^R)-v(s_g^R)} = \sqrt{2}e^{0.5v(s_0^R)} = \frac{0.5}{1 + \sqrt{N}}$$

$$q_1(s_0^R, a_2) = e^{B_0^v(s_0^R,a_2)} = e^{v(s_0^R)-v(s_1^R)} = \sqrt{2N}e^{0.5v(s_0^R)} = \frac{0.5\sqrt{N}}{1 + \sqrt{N}}$$

$$q_1(s_1^R, a) = e^{B_0^v(s_1^R,a)} = e^{v(s_1^R)} = \frac{1}{\sqrt{2N}}e^{0.5v(s_0^R)} = \frac{0.25}{\sqrt{N}(1 + \sqrt{N})}$$

$$q_1(s_g^R, a) = e^{B_0^v(s_g^R,a)} = e^{v(s_g^R)} = \frac{1}{\sqrt{2}}e^{0.5v(s_0^R)} = \frac{0.25}{1 + \sqrt{N}}.$$

**Let's now look at general** $k \geqslant 1$**:** Since $q_k$ is an occupancy measure, it satisfies the properties of the dynamics of the MDP (see the definition of $\Delta(T)$ in Section 2) and we have that for any $s \in \mathcal{S}$:
$\sum_{a\in\mathcal{A}} q_k(s, a) = \sum_{s'\in\mathcal{S}}\sum_{a'\in\mathcal{A}} p(s|s', a')q_k(s', a') + \mathbb{I}\{s = s_0\}$. In particular, this gives

- $s = s_0$: $q_k(s_0) = 1$
- $s = s_0^L$: $q_k(s_0^L) = 0.5q_k(s_0) = 0.5$

- $s = s_g^L$: $q_k(s_g^L) = q_k(s_0^L) = 0.5$
- $s = s_0^R$: $q_k(s_0^R) = 0.5 q_k(s_0) = 0.5$
- $s = s_g^R$: $q_k(s_g^R) = q_k(s_0^R, a_1)$
- $s = s_1^R$: $q_k(s_1^R) = \frac{1}{N} q_k(s_0^R, a_2)$

This leads to the following simplifications in the derivatives of $\mathcal{D}_k(v)$:

$$\frac{\partial \mathcal{D}_k(v)}{\partial v(s_0)} = e^{v(s_0) - 0.5v(s_0^L) - 0.5v(s_0^R)} - 1 = 0$$

$$\frac{\partial \mathcal{D}_k(v)}{\partial v(s_0^L)} = -0.5 + q_k(s_0^L, a_1) e^{v(s_0^L) - \eta c_k(s_0^L, a_1) - v(s_g^L)} + (1/2 - q_k(s_0^L, a_1)) e^{v(s_0^L) - \eta c_k(s_0^L, a_2) - v(s_g^L)} = 0$$

$$\frac{\partial \mathcal{D}_k(v)}{\partial v(s_g^L)} = 0.5 e^{v(s_g^L)} - 0.5 = 0 \implies e^{v(s_g^L)} = 1$$

$$\frac{\partial \mathcal{D}_k(v)}{\partial v(s_0^R)} = -0.5 + q_k(s_0^R, a_1) e^{v(s_0^R) - v(s_g^R)} + (1/2 - q_k(s_0^R, a_1)) e^{v(s_0^R) - \eta - v(s_1^R)} = 0$$

$$\frac{\partial \mathcal{D}_k(v)}{\partial v(s_1^R)} = -q_k(s_0^R, a_2) e^{v(s_0^R) - \eta - v(s_1^R)} + q_k(s_0^R, a_2) e^{v(s_1^R)} = 0 \implies e^{v(s_0^R)} = e^{\eta + 2v(s_1^R)}$$

$$\frac{\partial \mathcal{D}_k(v)}{\partial v(s_g^R)} = -q_k(s_0^R, a_1) e^{v(s_0^R) - v(s_g^R)} + q_k(s_0^R, a_1) e^{v(s_g^R)} = 0 \implies e^{v(s_0^R)} = e^{2v(s_g^R)}.$$

**Left part of the MDP:**

$$\frac{\partial \mathcal{D}_k(v)}{\partial v(s_0^L)} = 0 \implies q_k(s_0^L, a_1) e^{v(s_0^L) - \eta \frac{1 + (-1)^k}{2}} + (0.5 - q_k(s_0^L, a_1)) e^{v(s_0^L) - 0.5\eta} = 0.5$$

$$\implies e^{v(s_0^L)} = \frac{0.5}{q_k(s_0^L, a_1) e^{-\eta \frac{1 + (-1)^k}{2}} + (0.5 - q_k(s_0^L, a_1)) e^{-0.5\eta}}$$

$$k + 1 = 2 \implies e^{v(s_0^L)} = \frac{0.5}{0.25 + 0.25 e^{-0.5\eta}}$$

$$\implies q_2(s_0^L, a_1) = q_1(s_0^L, a_1) e^{B_1^v(s_0^L, a_1)} = 0.25 e^{v(s_0^L) - v(s_g^L)} = \frac{0.5}{1 + e^{-0.5\eta}}$$

$$k + 1 = 3 \implies e^{v(s_0^L)} = \frac{0.5}{q_1(s_0^L, a_1) e^{-\eta} + (0.5 - q_1(s_0^L, a_1)) e^{-0.5\eta}}$$

$$\implies e^{v(s_0^L)} = \frac{0.5}{\frac{0.5}{1 + e^{-0.5\eta}} e^{-\eta} + \frac{0.5 e^{-0.5\eta}}{1 + e^{-0.5\eta}} e^{-0.5\eta}} = 0.25 \frac{e^{\eta}}{q_2(s_0^L, a_1)}$$

$$\implies q_3(s_0^L, a_1) = q_2(s_0^L, a_1) e^{v(s_0^L) - \eta - v(s_g^L)} = 0.25$$

$$k \text{ even} \implies q_k(s_0^L, a_1) = \frac{0.5}{1 + e^{-0.5\eta}}$$

$$k \text{ odd} \implies q_k(s_0^L, a_1) = 0.25,$$

where the last two lines follow by a straightforward induction. Hence the losses suffered by OMD on the left part of the MDP are:

$$\sum_{k=1}^{K} \left\{ q_k(s_0^L, a_1) c_k(s_0^L, a_1) + q_k(s_0^L, a_2) c_k(s_0^L, a_2) \right\} = \sum_{k=1}^{K} \left\{ q_k(s_0^L, a_1) \cdot \frac{1 + (-1)^k}{2} + 0.5 \cdot (0.5 - q_k(s_0^L, a_1)) \right\}$$

$$= \sum_{k=1}^{K} \left\{ q_k(s_0^L, a_1) \cdot \frac{(-1)^k}{2} + 0.25 \right\}$$

$$= 0.25K + 0.5 \sum_{t=1}^{K/2} \left\{ q_{2t}(s_0^L, a_1) - q_{2t-1}(s_0^L, a_1) \right\}$$

$$= 0.25K + 0.5 \sum_{t=1}^{K/2} \left\{ \frac{0.5}{1 + e^{-0.5\eta}} - 0.25 \right\}$$

$$= 0.25K + 0.5 \frac{K}{2} \frac{0.5 - 0.25 - 0.25e^{-0.5\eta}}{1 + e^{-0.5\eta}}$$

$$= 0.25K + \frac{K}{16} \cdot \frac{1 - e^{-0.5\eta}}{1 + e^{-0.5\eta}}$$

$$= 0.25K + \frac{K}{16} \cdot \min\left\{\frac{\eta}{5}, \frac{1}{2}\right\}. \tag{7}$$

**Right part of the MDP:**

$$\frac{\partial \mathcal{D}_k(v)}{\partial v(s_0^R)} = 0 \implies e^{v(s_0^R)} = \frac{0.5}{q_k(s_0^R, a_1)e^{-v(s_g^R)} + (1/2 - q_k(s_0^R, a_1))e^{-\eta - v(s_1^R)}}$$

$$\frac{\partial \mathcal{D}_k(v)}{\partial v(s_1^R)} = 0 \implies v(s_0^R) = \eta + 2v(s_1^R)$$

$$\frac{\partial \mathcal{D}_k(v)}{\partial v(s_g^R)} = 0 \implies v(s_0^R) = v(s_g^R)$$

$$\implies e^{v(s_0^R)} = \frac{0.5}{q_k(s_0^R, a_1)e^{-0.5v(s_0^R)} + (1/2 - q_k(s_0^R, a_1))e^{-\eta - 0.5v(s_0^R) + 0.5\eta}}$$

$$\implies e^{0.5v(s_0^R)} = \frac{0.5}{q_k(s_0^R, a_1) + (1/2 - q_k(s_0^R, a_1))e^{-0.5\eta}} = \frac{0.5}{q_k(s_0^R, a_1) + q_k(s_0^R, a_2)e^{-0.5\eta}}$$

$$\implies q_{k+1}(s_0^R, a_1) = q_k(s_0^R, a_1)e^{B_k^v(s_0^R, a_1)} = q_k(s_0^R, a_1)e^{v(s_0^R) - v(s_g^R)} = q_k(s_0^R, a_1)e^{0.5v(s_0^R)}$$

$$\implies q_{k+1}(s_0^R, a_1) = 0.5 \frac{q_k(s_0^R, a_1)}{q_k(s_0^R, a_1) + (1/2 - q_k(s_0^R, a_1))e^{-0.5\eta}}$$

$$\implies \frac{q_{k+1}(s_0^R, a_1)}{q_{k+1}(s_0^R, a_2)} = \frac{q_{k+1}(s_0^R, a_1)}{(1/2 - q_{k+1}(s_0^R, a_1))} = \frac{q_k(s_0^R, a_1)}{(1/2 - q_k(s_0^R, a_1))e^{-0.5\eta}} = e^{0.5\eta} \frac{q_k(s_0^R, a_1)}{q_k(s_0^R, a_2)}$$

$$\implies \frac{q_{k+1}(s_0^R, a_1)}{q_{k+1}(s_0^R, a_2)} = e^{0.5k\eta} \frac{q_1(s_0^R, a_1)}{q_1(s_0^R, a_2)} = e^{0.5k\eta} \frac{0.5}{0.5\sqrt{N}} = \frac{1}{\sqrt{N}}e^{0.5k\eta}$$

$$\implies q_{k+1}(s_0^R, a_2) = \frac{0.5}{1 + \frac{1}{\sqrt{N}}e^{0.5k\eta}} = \frac{0.5\sqrt{N}}{\sqrt{N} + e^{0.5\eta k}}.$$

This also holds for $k = 0$ (as shown above). Hence, the losses suffered by OMD on the right part of the MDP are

$$\sum_{k=1}^{K} q_k(s_0^R, a_2)c_k(s_0^R, a_2) = \sum_{k=1}^{K} \frac{0.5\sqrt{N}}{\sqrt{N} + e^{0.5k\eta}}$$

$$\geqslant \int_1^{K+1} \frac{0.5\sqrt{N}}{\sqrt{N} + e^{0.5\eta x}}dx = 0.5K - \int_1^{K+1} \frac{0.5e^{0.5\eta x}}{\sqrt{N} + e^{0.5\eta x}}dx$$

$$= 0.5K - \left[\frac{1}{\eta}\log(\sqrt{N} + e^{0.5\eta x})\right]_1^{K+1}$$

$$= 0.5K - \frac{1}{\eta}\log(\sqrt{N} + e^{0.5\eta(K+1)}) + \frac{1}{\eta}\log(\sqrt{N} + e^{0.5\eta})$$

$$\geqslant 0.5K - \frac{1}{\eta}\log(2e^{0.5\eta(K+1)}) + \frac{1}{\eta}\log\sqrt{N} \qquad \text{assuming } \sqrt{N} \leqslant e^{0.5\eta(K+1)}$$

$$= 0.5K - 0.5(K+1) - \frac{1}{\eta}\log 2 + \frac{1}{2\eta}\log N$$

$$= -0.5 + \frac{1}{2\eta}\log\frac{N}{4}.$$

If $\sqrt{N} > e^{0.5\eta(K+1)}$, then we have

$$0.5K - \frac{1}{\eta}\log(\sqrt{N} + e^{0.5\eta(K+1)}) + \frac{1}{\eta}\log(\sqrt{N} + e^{0.5\eta}) = 0.5K + \frac{1}{\eta}\log\left(\frac{\sqrt{N} + e^{0.5\eta}}{\sqrt{N} + e^{0.5\eta(K+1)}}\right)$$

$$\geqslant 0.5K + \frac{1}{\eta}\log\left(\frac{e^{0.5\eta(K+1)} + e^{0.5\eta}}{2e^{0.5\eta(K+1)}}\right)$$

$$\geqslant 0.5K + \frac{1}{\eta}\log\left(\frac{1 + e^{-0.5\eta K}}{2}\right)$$

$$\geqslant 0.25K,$$

using that $\frac{1+e^{-0.5Kx}}{2} \geqslant e^{-0.25Kx}$ since $cosh(x) \geqslant 1$. So we have

$$\sum_{k=1}^{K} q_k(s_0^R, a_2)c_k(s_0^R, a_2) \geqslant \min\left\{-0.5 + \frac{1}{2\eta}\log\frac{N}{4}, 0.25K\right\}. \tag{8}$$

Combining the losses from the left part in (7) and from the right part in (8), we have:

$$\sum_{k=1}^{K}\langle q_k, c_k\rangle = \sum_{k=1}^{K}\left\{q_k(s_0^L, a_1)c_k(s_0^L, a_1) + q_k(s_0^L, a_2)c_k(s_0^L, a_2)\right\} + \sum_{k=1}^{K}\left\{q_k(s_0^R, a_2)c_k(s_0^R, a_2)\right\}$$

$$\geqslant 0.25K + \frac{K}{16}\cdot\min\left\{\frac{\eta}{5}, \frac{1}{2}\right\} + \min\left\{-0.5 + \frac{1}{2\eta}\log\frac{N}{4}, 0.25K\right\}.$$

**Regret lower-bound:** consider $q_\star$ defined as follows:

- $q_\star(s_0, a) = 1/2$ for all $a \in \mathcal{A}$
- $q_\star(s_0^L, a_2) = 1/2$, $q_\star(s_0^L, a_1) = 0$
- $q_\star(s_g^L, a) = 1/4$ for all $a \in \mathcal{A}$
- $q_\star(s_0^R, a_1) = 1/2$, $q_\star(s_0^R, a_2) = 0$, $q_\star(s_i^R, a) = 0$
- $q_\star(s_g^R, a) = 1/4$ for all $a \in \mathcal{A}$

It is straightforward to check that $q_\star$ satisfies the flow constraints and is an occupancy measure. We obtain

$$\sum_{k=1}^{K}\langle q_\star, c_k\rangle = \sum_{k=1}^{K}\left\{q_\star(s_0^L, a_2)\cdot 0.5\right\} = 0.25K$$

$$\implies R_K \geqslant \sum_{k=1}^{K}\langle q_k - q_\star, c_k\rangle \geqslant \frac{K}{16}\cdot\min\left\{\frac{\eta}{5}, \frac{1}{2}\right\} + \min\left\{-0.5 + \frac{1}{2\eta}\log\frac{N}{4}, 0.25K\right\}$$

$$\geqslant \min\left\{\frac{1}{2}\sqrt{\frac{1}{10}K\log\frac{N}{4}}, \frac{K}{32}\right\} - 0.5.$$

Recalling that $N = S - 5$, we have $R_K = \Omega\left(\min\{\sqrt{K\log S}, K\}\right)$ for an MDP where the sparsity level is $M = 3$, concluding the proof. $\qquad\square$

# B Efficient implementation of OMD using our regularizer

In this section, we describe how the OMD update with our regularizer from Section 4 defined in (4) can be computed efficiently. This closely follows Appendix B.1 of [28], who provide a similar description for the negative entropy.

Recall the regularizer $\psi_p(q) = p \sum_{s \in \mathcal{S}} \sum_{a \in \mathcal{A}} q(s,a)^{1+1/p} - p$ for $q \in \mathbb{R}_{\geq 0}^{\Gamma}$. We have

$$\nabla \psi_p(q) = (p+1) \cdot q(s,a)^{1/p}.$$

The Bregman divergence is defined as:

$$
\begin{aligned}
D_{\psi_p}(q,q') &= \psi_p(q) - \psi_p(q') - \langle \nabla \psi_p(q'), q - q' \rangle \\
&= \sum_{s \in \mathcal{S}} \sum_{a \in \mathcal{A}} \left\{ p \cdot q(s,a)^{1+1/p} - p \cdot q'(s,a)^{1+1/p} \right\} \\
&\quad - (p+1) \sum_{s \in \mathcal{S}} \sum_{a \in \mathcal{A}} \left\{ q'(s,a)^{1/p} q(s,a) - q'(s,a)^{1+1/p} \right\} \\
&= \sum_{s \in \mathcal{S}} \sum_{a \in \mathcal{A}} \left\{ q'(s,a)^{1+1/p} + q(s,a) \cdot \left[ p \cdot q(s,a)^{1/p} - (p+1) \cdot q'(s,a)^{1/p} \right] \right\}.
\end{aligned}
$$

Recall that OMD with the above regularizer computes the occupancy measures as follows - see (1):

$$q_1 = \arg \min_{q \in \Delta(T)} \psi_p(q), \qquad q_{k+1} = \arg \min_{q \in \Delta(T)} \left\{ \eta \cdot \langle q, c_k \rangle + D_{\psi_p}(q, q_k) \right\}.$$

As shown in [24] (Theorem 6.15), each of these steps can be split into an unconstrained minimization step, and a projection step. Thus, $q_1$ can be computed as follows

$$q_1' = \arg \min_{q \in \mathbb{R}_{\geq 0}^{\Gamma}} \psi_p(q)$$

$$q_1 = \arg \min_{q \in \Delta(T)} D_{\psi_p}(q, q_1'),$$

where $q_1'$ has a closed-from solution $q_1'(s,a) = 1$ for every $s \in \mathcal{S}$ and $a \in \mathcal{A}$. Similarly, $q_{k+1}$ is computed as follows for every $k = 1, ..., K-1$:

$$q_{k+1}' = \arg \min_{q \in \mathbb{R}_{\geq 0}^{\Gamma}} \left\{ \eta \cdot \langle q, c_k \rangle + D_{\psi_p}(q, q_k) \right\}$$

$$q_{k+1} = \arg \min_{q \in \Delta(T)} D_{\psi_p}(q, q_{k+1}'),$$

where again $q_{k+1}'$ has a closed-from solution $q_{k+1}'(s,a) = \left[ q_k(s,a)^{1/p} - \frac{\eta}{p+1} c_k(s,a) \right]_+^p$ for every $s \in \mathcal{S}$ and $a \in \mathcal{A}$ (follows from straightforwardly differentiating above objective and setting to 0 and accounting for the non-negativity of occupancy measures) - we use notation $a_+ = \max\{0, a\}$.

For the projection step, we start by formulating it as a constrained convex optimization problem:

$$
\begin{aligned}
\min_{q \in \mathbb{R}^{\Gamma}} D_{\psi_p}(q, q_{k+1}') \quad \text{s.t.} \quad & \sum_{a \in \mathcal{A}} q(s,a) - \sum_{s' \in \mathcal{S}} \sum_{a' \in \mathcal{A}} P(s'|s,a) q(s',a') = \mathbb{I}\{s = s_0\} \qquad \forall s \in \mathcal{S} \\
& \sum_{s \in \mathcal{S}} \sum_{a \in \mathcal{A}} q(s,a) \leq T \\
& q(s,a) \geq 0 \qquad \forall (s,a) \in \mathcal{S} \times \mathcal{A}.
\end{aligned}
$$

The problem can be solved by considering the Lagrangian with Lagrange multipliers $\lambda$ and $\{v(s)\}_{s \in \mathcal{S}}$:

$$
\begin{aligned}
\mathcal{L}(q, \lambda, v) &= D_{\psi_p}(q, q_{k+1}') + \lambda \left( \sum_{s \in \mathcal{S}} \sum_{a \in \mathcal{A}} q(s,a) - T \right) + \sum_s v(s) \left( \sum_{s',a'} P(s|s',a') q(s',a') + \mathbb{I}\{s = s_0\} - \sum_a q(s,a) \right) \\
&= D_{\psi_p}(q, q_{k+1}') + \sum_{s \in \mathcal{S}} \sum_{a \in \mathcal{A}} q(s,a) \left( \lambda + \sum_{s' \in \mathcal{S}} P(s'|s,a) v(s') - v(s) \right) + v(s_0) - \lambda T,
\end{aligned}
$$

Differentiating the Lagrangian with respect to any $q(s,a)$ and setting to 0, we get

$$\frac{\partial \mathcal{L}(q,\lambda,v)}{\partial q(s,a)} = \nabla \psi_p(q)(s,a) - \nabla \psi_p(q'_{k+1})(s,a) + \lambda + \sum_{s'\in\mathcal{S}} P(s'|s,a)v(s') - v(s)$$

$$= (p+1)q(s,a)^{1/p} - (p+1)q'_{k+1}(s,a)^{1/p} + \lambda + \sum_{s'\in\mathcal{S}} P(s'|s,a)v(s') - v(s) = 0.$$

$$\implies q_{k+1}(s,a) = \left[ q'_{k+1}(s,a)^{1/p} - \frac{\lambda + \sum_{s'\in\mathcal{S}} P(s'|s,a)v(s') - v(s)}{p+1} \right]_+^p.$$

This formula is also valid for $k=0$ by setting $c_0(s,a) = 0$ and $q_0(s,a) = 1$ for every $s \in \mathcal{S}$ and $a \in \mathcal{A}$.

To compute the value of $\lambda$ and $v$ at the optimum, we write the dual problem $\mathcal{D}(\lambda,v) = \min_q \mathcal{L}(q,\lambda,v)$ by substituting $q_{k+1}$ back into $\mathcal{L}$:

$$\mathcal{D}(\lambda,v) = D_{\psi_p}(q_{k+1}, q'_{k+1}) + \sum_{s\in\mathcal{S}}\sum_{a\in\mathcal{A}} q_{k+1}(s,a)\left(\lambda + \sum_{s'\in\mathcal{S}} P(s'|s,a)v(s') - v(s)\right) + v(s_0) - \lambda T.$$

Recall that $q'_{k+1}(s,a) = \left[ q_k(s,a)^{1/p} - \frac{\eta}{p+1}c_k(s,a) \right]^p$, so (ignoring terms independent of $\lambda, v$, e.g. $q'_{k+1}(s,a)$):

$$D_{\psi_p}(q_{k+1}, q'_{k+1}) = \sum_{s\in\mathcal{S}}\sum_{a\in\mathcal{A}} \left\{ q'_{k+1}(s,a)^{1+1/p} + q_{k+1}(s,a)\cdot\left[ pq_{k+1}(s,a)^{1/p} - (p+1)q'_{k+1}(s,a)^{1/p} \right] \right\}$$

$$\propto \sum_{s\in\mathcal{S}}\sum_{a\in\mathcal{A}} \left\{ q_{k+1}(s,a)\cdot\left[ p\left(q'_{k+1}(s,a)^{1/p} - \frac{\lambda + \sum_{s'\in\mathcal{S}} P(s'|s,a)v(s') - v(s)}{p+1}\right)_+ \right.\right.$$

$$\left.\left. - (p+1)q'_{k+1}(s,a)^{1/p} \right] \right\}$$

$$= \sum_{s\in\mathcal{S}}\sum_{a\in\mathcal{A}} \left\{ q_{k+1}(s,a)\cdot\left[ p\left(q'_{k+1}(s,a)^{1/p} - \frac{\lambda + \sum_{s'\in\mathcal{S}} P(s'|s,a)v(s') - v(s)}{p+1}\right) \right.\right.$$

$$\left.\left. - (p+1)q'_{k+1}(s,a)^{1/p} \right] \right\} \quad \text{since if } q_{k+1}(s,a) = 0, \text{ then the whole term is } 0$$

$$= \sum_{s\in\mathcal{S}}\sum_{a\in\mathcal{A}} \left\{ q_{k+1}(s,a)\cdot\left[ -q'_{k+1}(s,a)^{1/p} - \frac{p}{p+1}\left(\lambda + \sum_{s'\in\mathcal{S}} P(s'|s,a)v(s') - v(s)\right) \right] \right\}$$

$$\implies \mathcal{D}(\lambda,v) \propto \sum_{s\in\mathcal{S}}\sum_{a\in\mathcal{A}} \left\{ q_{k+1}(s,a)\cdot\left[ -q'_{k+1}(s,a)^{1/p} - \frac{p}{p+1}\left(\lambda + \sum_{s'\in\mathcal{S}} P(s'|s,a)v(s') - v(s)\right) \right.\right.$$

$$\left.\left. + \left(\lambda + \sum_{s'\in\mathcal{S}} P(s'|s,a)v(s') - v(s)\right) \right] \right\} + v(s_0) - \lambda T$$

$$= \sum_{s\in\mathcal{S}}\sum_{a\in\mathcal{A}} \left\{ q_{k+1}(s,a)\cdot\left[ -q'_{k+1}(s,a)^{1/p} + \frac{\lambda + \sum_{s'\in\mathcal{S}} P(s'|s,a)v(s') - v(s)}{p+1} \right] \right\} + v(s_0) - \lambda T$$

$$= -\sum_{s\in\mathcal{S}}\sum_{a\in\mathcal{A}} q_{k+1}(s,a)^{1+1/p} + v(s_0) - \lambda T$$

$$= -\sum_{s\in\mathcal{S}}\sum_{a\in\mathcal{A}}\left[ q'_{k+1}(s,a)^{1/p} - \frac{\lambda + \sum_{s'\in\mathcal{S}} P(s'|s,a)v(s') - v(s)}{p+1} \right]_+^{1+p} + v(s_0) - \lambda T.$$

Maximizing the dual gives $\lambda$ and $v$ or equivalently, we can minimize the negation of the dual:

$$\lambda_{k+1}, v_{k+1} = \arg\min_{\lambda\geq 0, v} \sum_{s\in\mathcal{S}}\sum_{a\in\mathcal{A}}\left[ q'_{k+1}(s,a)^{1/p} - \frac{\lambda + \sum_{s'\in\mathcal{S}} P(s'|s,a)v(s') - v(s)}{p+1} \right]_+^{1+p} - v(s_0) + \lambda T.$$

This is a convex optimization problem subject only to non-negativity constraints, and can be efficiently solved using iterative methods ( e.g. gradient descent).

# C   The benefits of sparsity - upper bounds

We restate the main theorem proved in Section 4.

**Theorem 4.1.** *Consider OMD with $\psi_p$ as regularizer. If $T > e$ is such that $q_{\pi^\star} \in \Delta(T)$, $\eta = \sqrt{\frac{pT^{1+1/p}}{KDM^{1/p}}}$, $p = \log(TM)$, then $\mathbb{E}\left[R_K\right] \leqslant \mathcal{O}\left(\sqrt{DKT\log(MT)}\right)$.*

We include the missing details from the proof given in Section 4. First, recall from (4) that

$$\psi_p(q) = p \cdot \left(-1 + \|q\|_{1+1/p}^{1+1/p}\right) = p \cdot \left(-1 + \sum_{s \in \mathcal{S}} \sum_{a \in \mathcal{A}} |q(s,a)|^{1+1/p}\right)$$

$$\implies \frac{\partial \psi_p(q)}{\partial q(s,a)} = (p+1)q(s,a)^{1/p}$$

$$\implies \frac{\partial^2 \psi_p(q)}{\partial q(s,a)^2} = \left(1 + \frac{1}{p}\right)q(s,a)^{1/p-1}$$

$$\implies \nabla \psi_p(q) = (p+1)q^{1/p}, \qquad \nabla^2 \psi_p(q) = \operatorname{diag}\left(\frac{p+1}{p}q^{1/p-1}\right)$$

We implicitly assumed here that $\psi_p$ is defined on $\mathbb{R}_{\geqslant 0}^{\Gamma}$. The missing details are:

- $\psi_p$ satisfies the condition (2) with $\alpha = 1$:

$$\nabla \psi_p(q) \in \left[\nabla \psi_p(q_k), \nabla \psi_p(q_k) - \eta c_k\right] \implies q^{1/p}(s,a) \leqslant q_k^{1/p}(s,a)$$

$$\implies \frac{1}{q(s,a)} \geqslant \frac{1}{q_k(s,a)}$$

$$\implies \frac{1}{q^{1-1/p}(s,a)} \geqslant \frac{1}{q_k^{1-1/p}(s,a)}$$

$$\implies \nabla^2 \psi_p(q) \succeq \nabla^2 \psi_p(q_k).$$

- $\nabla^2 \psi_p(q)^{-1} = \operatorname{diag}\left(\frac{p}{p+1}q^{1-1/p}\right)$: follows directly from the expression for $\nabla^2 \psi_p(q)$ above.

We now turn to the description of the parameter-free algorithm and the proof of its corresponding regret bound (Theorem 4.4).

## C.1   Sparse-agnostic bound

For the unknown sparsity level, we use the same approach as in [19], dividing the episode horizon into segments, where each segment will run OMD from scratch with an increasing sparsity level guess. Crucially, there will be at most $O(\log \log M)$ such segments.

Define

- $M = \max_{k \in [K]} \|c_k\|_0$, the true sparsity level across the horizon.

- $B = \lceil \log_2 \log_2 M \rceil$, the maximum number of segments.

- $m(b) = 2^{2^b}$, the assumed sparsity level during the $b$-th segment (or interval $I(b)$ below). The reason to use a double exponential is that this sparse-agnostic procedure brings an extra $B$ factor to the regret bound: if we use $2^b$ then $B = O(\log M)$ which harms the regret bound.

- for $1 \leqslant b \leqslant B$, $\tau(b) = \min\{1 \leqslant k \leqslant K \mid \|c_k\|_0 > m(b)\}$, the first episode in which the sparsity level of the loss vector exceeds $m(b)$. We also define $\tau(0) = 0$ and $\tau(B) = K$.

Using this notation, we can partition the horizon $[K]$ as intervals $(I(b))_{b \in [B]}$ according to the episodes $\tau(b)$ where the thresholds $m(b)$ are first exceeded. For $1 \leqslant b \leqslant B$:

$$I(b) = \begin{cases} [\tau(b-1) + 1, \tau(b)] & \text{if } \tau(b-1) < \tau(b) \\ \varnothing & \text{if } \tau(b-1) = \tau(b) \end{cases}$$

Let $b_k = \min\{b \geqslant 1 \mid \tau(b) \geqslant k\}$ be the index of the only interval to which episode $k$ belongs.

Now we define the OMD parameters used in interval $I(b)$, in which we essentially use the parameters from Theorem 4.1 assuming the sparsity level is $m(b)$:

---

**Algorithm 1** Sparse-Agnostic Mirror Descent

---

**Input**: $T, K, D$
**Initialize:** $p \leftarrow \log 2^2 T$, $\eta \leftarrow \sqrt{T^{1+1/p}/(pDK2^{2/p})}$, $b \leftarrow 1$, $q_1 \leftarrow \arg\min_{q \in \Delta(T)} \psi_p(q)$
**for** $k = 1, \dots, K$ **do**
    Play $q_k$ and Observe $c_k$
    **if** $\|c_k\|_0 \leqslant 2^{2^b}$ **then**
        $q_{k+1} = \arg\min_{q \in \Delta_T} \langle q, c_k \rangle + D_{\psi_p}(q, q_k)$
    **else**
        $b \leftarrow \lceil \log_2 \log_2 \|c_k\|_0 \rceil$
        $p \leftarrow \log 2^{2^b} T$
        $\eta \leftarrow \sqrt{pT^{1+1/p}/(DK2^{2/p})}$
        $q_{k+1} \leftarrow \arg\min_{q \in \Delta(T)} \psi_p(q)$
    **end if**
**end for**

---

- the parameter of our regularizer is $p(b) = \log(m(b)T)$.

- the step-size is $\eta(b) = \sqrt{\frac{p(b)T^{1+1/p(b)}}{DKm(b)^{1/p(b)}}}$.

Recall that our regularizer with parameter $p$ is given by $\psi_p(q) = p\left(-1 + \|q\|_{1+1/p}^{1+1/p}\right)$. At episode $k$, we use the parameter $p(b_k)$ defined above, i.e. using the index value $b_k$ of the interval $I(b_k)$ to which episode $k$ belongs to. The OMD update is then defined by:

$$q_k = \nabla \psi_{p(b_k)}^\star \left( \eta(b_k) \sum_{k' < k,\, k' \in I(b_k)} c_{k'} \right), \; k = 1, \dots, K\,.$$

The full procedure is given in Algorithm 1. The following lemma shows the cost of being sparse-agnostic is an additive $T$ term and a double-logarithmic factor in the sparsity level $M$.

**Lemma C.1.** *Consider running Algorithm 1. If $T > e$ is such that $q_{\pi^\star} \in \Delta(T)$, then $\mathbb{E}\left[R_K\right] \leqslant \mathcal{O}\left(TB + B\sqrt{DKT\log(MT)}\right)$.*

*Proof.* Fix (interval) $b \in [B]$. On the time interval $I(b)$, we run OMD with regularizer $\psi_{p(b)}$, learning rate $\eta(b)$ and we consider the (expected) interval regret $R(b) = \sum_{k \in I(b)}\langle q_{\pi_k}, c_k \rangle - \min_{\pi \in \Pi_p} \sum_{k \in I(b)}\langle q_{\pi^\star}, c_k \rangle$. Crucially we know that up to the last time step of the interval, we have a bound $m(b)$ on the sparsity for all rounds but the last.

Since $J_k^\pi \leqslant T$ for any $k$, we just consider the regret on the rounds not including $\tau(b)$ for which we suffer a regret of at most $T$:

$$R(b) = \sum_{k \in I(b)} \langle q_{\pi_k} - q_{\pi^\star}, c_k \rangle \leqslant T + \sum_{\substack{k \in I(b) \\ k < \tau(b)}} \langle q_{\pi_k} - q_{\pi^\star}, c_k \rangle.$$

For the other rounds we follow similar steps as in the proof of Theorem 4.1:

$$\sum_{\substack{k \in I(b) \\ k < \tau(b)}} \langle q_{\pi_k} - q_{\pi^\star}, c_k \rangle \leqslant \frac{p(b)T^{1+1/p(b)}}{\eta(b)} + \frac{\eta(b)}{2} \sum_{\substack{k \in I(b) \\ k < \tau(b)}} \|c_k\|_1^{1/p(b)} (\langle c_k, q_k \rangle + 1)$$

$$\leqslant \frac{p(b)T^{1+1/p(b)}}{\eta(b)} + \frac{\eta(b)m(b)^{1/p(b)}}{2} \sum_{\substack{k \in I(b) \\ k < \tau(b)}} (\langle c_k, q_k \rangle + 1)$$

$$\implies \sum_{\substack{k \in I(b) \\ k < \tau(b)}} \langle q_{\pi_k} - q_{\pi^\star}, c_k \rangle \leqslant \frac{1}{1 - \eta(b)m(b)^{1/p(b)}} \left[ \frac{p(b)T^{1+1/p(b)}}{\eta(b)} + \frac{\eta(b)m(b)^{1/p(b)}}{2} \sum_{\substack{k \in I(b) \\ k < \tau(b)}} (\langle c_k, q_{\pi^\star} \rangle + 1) \right]$$

$$\leqslant \frac{2p(b)T^{1+1/p(b)}}{\eta(b)} + \eta(b)m(b)^{1/p(b)} \sum_{\substack{k\in I(b) \\ k<\tau(b)}} (\langle c_k, q_{\pi^\star}\rangle + 1)$$

$$\leqslant \frac{2p(b)T^{1+1/p(b)}}{\eta(b)} + \eta(b)m(b)^{1/p(b)} \sum_{k\in I(b)} (\langle c_k, q_{\pi^\star}\rangle + 1),$$

where we used that $\frac{1}{1-\eta(b)m(b)^{1/p(b)}} \leqslant 2$ which is the case if $K > 8eT\log(MT)$ using how we defined $\eta(b)$. Using $\eta(b) = \sqrt{\frac{p(b)T^{1+1/p(b)}}{DKm(b)^{1/p(b)}}}$, we have

$$R(b) \leqslant T + \sqrt{p(b)T^{1+1/p(b)}DKm(b)^{1/p(b)}} \cdot \left(2 + \frac{\sum_{k\in I(b)}(\langle c_k, q_{\pi^\star}\rangle + 1)}{DK}\right)$$

$$\leqslant T + \sqrt{TDKe\log\big(m(b)T\big)} \cdot \left(2 + \frac{\sum_{k\in I(b)}(\langle c_k, q_{\pi^\star}\rangle + 1)}{DK}\right)$$

$$\leqslant T + \sqrt{2TDKe\log(MT)} \cdot \left(2 + \frac{\sum_{k\in I(b)}(\langle c_k, q_{\pi^\star}\rangle + 1)}{DK}\right),$$

where we used that $p(b) = \log\big(m(b)T\big)$ and that

$$b \leqslant B \leqslant 1 + \log_2\log_2 M \implies m(b) = 2^{2^b} \leqslant 2^{2^{1+\log_2\log_2 M}} = 2^{2\cdot\log_2 M} = M^2$$

$$\implies \log\big(m(b)T\big) = \log\big(M^2 T\big) \leqslant 2\log(MT).$$

Then

$$E[R_K] \leqslant \sum_{b=1}^{B} R(b)$$

$$\leqslant TB + \sqrt{2TDKe\log(MT)} \cdot \left(2B + \sum_{b=1}^{B}\frac{\sum_{k\in I(b)}(\langle c_k, q_{\pi^\star}\rangle + 1)}{DK}\right)$$

$$= TB + \sqrt{2TDKe\log(MT)} \cdot \left(2B + \frac{1}{D} + \frac{1}{DK}\sum_{k=1}^{K}\langle c_k, q_{\pi^\star}\rangle\right)$$

$$\leqslant TB + 4B\sqrt{2TDKe\log(MT)},$$

where the last step uses that $\sum_{k=1}^{K}\langle q_{\pi^\star}, c_k\rangle \leqslant \sum_{k=1}^{K}\langle q_{\pi_f}, c_k\rangle \leqslant K\|q_{\pi_f}\|_1\|c_k\|_\infty \leqslant DK$ and $D \geqslant 1$. □

### C.2 Fully parameter-free bound

We now turn our attention to the unknown hitting time of the optimal policy $T_\star$, where we can exploit the same technique presented in [8].

We run $N \approx \log K$ instances of Algorithm 1 where the $j$-th instance will set its parameter $T$ as $b(j)$ which is roughly $2^j$, so that there always exists an instance $j_\star$ such that $b(j_\star)$ is close to the unknown $T_\star$. Specifically, we run a scale invariant meta algorithm with a correction term as in [8] to obtain the desired bound (details in Algorithm 2).

**Theorem 4.4.** *If $K > \max\big(T_\star, \frac{T_\star}{D}\log(T_\star M)\big)$ and $T_\star > e$, Algorithm 2 guarantees $\mathbb{E}\left[R_K\right] \leqslant \tilde{O}\big(\sqrt{DKT_\star\log(MT_\star)} + T_\star\big)$, where the notation $\tilde{O}$ hides double-logarithmic factors.*

*Proof.* We closely follow the steps of the proof of Theorem 2 in [8]. We have

- $b(1) \geqslant T^{\pi_f}(s_0)$ so for all instances $\Delta(b(j))$ is non-empty and the instance is well-defined.

- Let $j^\star$ be the index of the instance with smallest $b(j^\star)$ that is larger than $T^\star$, i.e. $\frac{b(j^\star)}{2} \leqslant T_\star \leqslant b(j^\star)$. This instance exists since $b(N) \geqslant K > T_\star$.

**Algorithm 2** Fully Parameter-Free Online Mirror Descent for Sparse SSPs

---

**Define** $j_0 = \lceil \log_2 T^{\pi_f}(s_0) \rceil - 1, b(j) = 2^{j_0+j}, N = \lceil \log_2 K \rceil - j_0, \eta_j = \left( \sqrt{Db(j)K \log(b(j)M)} \right)^{-1/2}$

**Define** $\psi_p(p) = \sum_{j=1}^{N} \frac{1}{\eta_j} p(j) \log p(j)$

**Initialize:** $p_1 \in \Delta_N$, such that $p_1(j) = \frac{\eta_j}{\eta_1 N}, \forall j \neq 1$

**Initialize:** $N$ instances of Algorithm 1, with $j-$th instance $T = b(j)$

**for** $k = 1, \ldots, K$ **do**

    Obtain occupancy measures $q_k^j$ for $j \in [N]$

    Sample $j_k \sim p_k$, execute policy induced by $q_k^{j_k}$

    Receive $c_k$ and send it to all instances.

    Compute $\ell_k(j) = \langle q_k^j, c_k \rangle, a_k(j) = 4\eta_j \ell_k^2(j)$

    Update $p_{k+1} = \arg\min_{p \in \Delta_N} \langle p, \ell_k + a_k \rangle + D_{\psi_p}(p, p_k)$

**end for**

---

We start by decomposing the regret into the regret of the meta algorithm *w.r.t.* finding $j_\star$ and the regret of the $j_\star$ instance *w.r.t.* the best policy:

$$
\begin{aligned}
\mathbb{E}[R_K] &= \sum_{k=1}^{K} \sum_{j=1}^{N} p_k(j) \langle q_k^j, c_k \rangle - \sum_{k=1}^{K} \langle q_{\pi^\star}, c_k \rangle \\
&= \sum_{k=1}^{K} \sum_{j=1}^{N} p_k(j) \langle q_k^j, c_k \rangle - \langle q_k^{j_\star}, c_k \rangle + \sum_{k=1}^{K} \langle q_k^{j_\star} - q_{\pi^\star}, c_k \rangle \\
&= \underbrace{\sum_{k=1}^{K} \langle p_k - e_{j_\star}, \ell_k \rangle}_{\text{Meta-Regret}} + \underbrace{\sum_{k=1}^{K} \langle q_k^{j_\star} - q_{\pi^\star}, c_k \rangle}_{j_\star - \text{Regret}}
\end{aligned}
$$

where we consider $p_k, \ell_k$ as $N$-dimensional vectors and $e_{j_\star}$ as the basis vector with the $j_\star$ coordinate equal to 1.

By Lemma C.1 the $j_\star$-Regret is bounded by $\mathcal{O}\left(Bb(j_\star) + B\sqrt{Db(j_\star)K \log(b(j_\star)M)}\right) = \mathcal{O}\left(BT_\star + B\sqrt{DT_\star K \log(T_\star M)}\right)$.

This also allows to say that:

$$
\begin{aligned}
\sum_{k=1}^{K} \langle q_k^{j_\star}, c_k \rangle &\leq \sum_{k=1}^{K} \langle q_{\pi^\star}, c_k \rangle + \mathcal{O}\left(BT_\star + B\sqrt{DT_\star K \log(T_\star M)}\right) \\
&\leq \sum_{k=1}^{K} \langle q_{\pi_f}, c_k \rangle + \mathcal{O}\left(BT_\star + B\sqrt{DT_\star K \log(T_\star M)}\right) \\
&\leq DK + \mathcal{O}\left(BT_\star + B\sqrt{DT_\star K \log(T_\star M)}\right),
\end{aligned}
$$

which we will make use of just below.

For the meta algorithm regret, we can use Lemma 12 of [8] which guarantees that:

$$
\begin{aligned}
\mathbb{E}\left[\sum_{k=1}^{K} \langle p_k - e_{j_\star}, \ell_k \rangle\right] &= \mathcal{O}\left(\frac{2 + \log\left(N\sqrt{\frac{b(j_\star)}{b(1)}}\right)}{\eta_{j_\star}} + 4\eta_{j_\star} b(j_\star) \sum_{k=1}^{K} \langle q_k^{j_\star}, c_k \rangle\right) \\
&= \mathcal{O}\left(\frac{2 + \log\left(N\sqrt{\frac{b(j_\star)}{b(1)}}\right)}{\eta_{j_\star}} + 4\eta_{j_\star} b(j_\star) \left(DK + BT_\star + B\sqrt{DT_\star K \log(T_\star M)}\right)\right)
\end{aligned}
$$

$$= \tilde{\mathcal{O}}\left(\frac{\log T_\star}{\eta_{j_\star}} + \eta_{j_\star} T_\star \left(DK + \sqrt{DT_\star K \log(T_\star M)}\right)\right)$$

$$= \tilde{\mathcal{O}}\left(\frac{\log T_\star}{\eta_{j_\star}} + \eta_{j_\star} DT_\star K\right)$$

$$= \tilde{\mathcal{O}}\left(\sqrt{DT_\star K \log T_\star}\right),$$

where we used

- the notation $\tilde{\mathcal{O}}$ to ignore *double*-logarithmic factors.
- $K \geqslant \frac{T_\star}{D} \log(T_\star M)$.
- $\eta_{j_\star} = \sqrt{\frac{\log T_\star}{DT_\star K}}$.
- $D \geqslant 1$.

Combining everything gives the result. $\qquad\square$

# D   State level sparsity

In this section, we consider a different notion of state-level sparsity:

$$M' = \max_k \max_{s \in \mathcal{S}} \sum_{a \in \mathcal{A}} \mathbb{I}\{c_k(s, a) > 0\} \leqslant A.$$

From Theorem 4 in [38], we have a cumulative loss bound for a version of OMD with negative entropy regularisation:

$$\mathbb{E}[R_K] \leqslant \tilde{\mathcal{O}}\Big(\sqrt{T_\star \sum_{k=1}^{K} J_k^{\pi_\star}}\Big),$$

from which we can exploit the state-level sparsity:

$$
\begin{aligned}
\sum_{k=1}^{K} J_k^{\pi^\star} &= \min_\pi \sum_{k=1}^{K} \mathbb{E}\Big[ \sum_{t=1}^{I_\pi(s_0)} c_k(s^t, a^t) | P, \pi, s^1 = s_0 \Big] \\
&\leqslant \sum_{k=1}^{K} \mathbb{E}\Big[ \sum_{t=1}^{I_{\pi_u}(s_0)} c_k(s^t, a^t) | P, \pi_u, s^1 = s_0 \Big] \\
&= \sum_{k=1}^{K} \mathbb{E}\Big[ \sum_{t=1}^{I_{\pi_u}(s_0)} \frac{1}{A} \sum_{a \in \mathcal{A}} c_k(s^t, a) | P, \pi_u, s^1 = s_0 \Big] \\
&\leqslant \frac{KM'}{A} \mathbb{E}\big[ I_{\pi_u}(s_0) | P, \pi_u, s^1 = s_0 \big] \\
&= \frac{KM'}{A} T^u(s_0),
\end{aligned}
$$

where $\pi_u$ is the uniform policy ($\pi_u(a|s) = 1/A$) and $T^u$ is its corresponding hitting-time. This gives the following regret bound:

$$\mathbb{E}[R_K] \leqslant \tilde{\mathcal{O}}\Big(\sqrt{KT_\star \frac{M'}{A} T^u(s_0)}\Big).$$

We can actually relate this result back to our original notion of sparsity since we know that $M' \leqslant M$. If $M \leqslant A$, then we can non-trivially bound $M'$ by $M$ and achieve a regret of $\tilde{\mathcal{O}}\Big(\sqrt{KT_\star \frac{M}{A} T^u(s_0)}\Big)$.

This result highlights that it is possible to achieve polynomial improvements from sparsity if we consider state-level sparsity $M'$ or $M < A$. However, this comes at the cost of a $T^u(s_0)$ factor. In the worst case, this additional factor will cancel the polynomial improvement. It could be an interesting avenue of future research to understand specific structural properties of the MDP that may lead to real polynomial improvements.

In the experts setting ($S = 1$), we have $M = M'$ and $T^u(s_0) = 1$ and this bound recovers the $\frac{M}{A}$ improvement of the expert setting. This provides some further insights into the performance of OMD with negative entropy regularisation and that in particular issues arise when there is at least 1 state with non-sparse costs even though most other states may have sparse costs.

# E    The benefits of sparsity - lower bound under sparsity

In this appendix, we prove our sparse lower bound result from Section 4.2. We first restate the result.

**Theorem 4.6.** *For any $D, T_\star, K, S, A$ with $T^\star \geqslant D \geqslant 3 \log S$, $S(A-1) \geqslant 400$, $K \geqslant \frac{800 T^\star}{D} \log M$ and $M \geqslant 101$, there exists an SSP instance with stochastic $M$-sparse costs, $S$ states and $A$ actions such that its diameter is $D$, the expected hitting time of the optimal policy is $T^\star$, and the expected regret with respect to the randomness of the losses for any learner after $K$ episodes is $\mathbb{E}[R_K] \geqslant \Omega\big(\sqrt{KT^\star D \log M}\big)$.*

*Proof.* Fix $B = \lceil \frac{\log S/2}{\log 2} \rceil - 2$. Fix $N = 2^{B+1} \geqslant 2^{\frac{\log S/2}{\log 2} - 1} = \frac{S}{4}$. Fix $L = \min\big\{ M - 1, N \cdot (A - 1)\big\} \geqslant \frac{M}{8}$. Fix $D' = D - B - 2$, $T' = T_\star - B - 1$, with $T_\star \geqslant D$.

We first describe the SSP instance with stochastic costs. Consider the following MDP $\mathcal{M} = (\mathcal{S}, \mathcal{A}, p, s_0, g)$ illustrated in Figure 4 and that we formally define below.

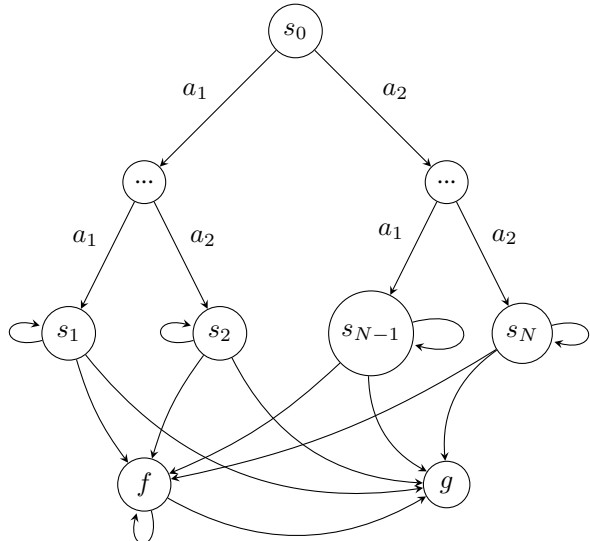

Figure 4: Diagram illustrating MDP construction for the proof of Theorem 4.6. Details are given below.

The first part of the states are represented by a binary tree of depth $B + 2$ and allow us to formerly consider the $N$ states at the bottom of the tree that matter, while avoiding an assumption on the existence of a state with $A \approx S$ actions as was done in prior work [8]. Each non-leaf node corresponds to a state with two actions transitioning (deterministically) to the left or right child respectively. The total number of nodes in the tree is

$$\sum_{i=0}^{B+1} 2^i = 2^{B+2} - 1 \leqslant 2^{\frac{\log S/2}{\log 2} + 1} - 1 = 2\frac{S}{2} - 1 = S - 1.$$

The total number of leaf nodes is $N = 2^{B+1} \geqslant \frac{S}{4}$. Denote the set of states corresponding to the leaf nodes by $\mathcal{S}_\ell = \big\{s_1, ..., s_N\big\}$. The root node is $s_0$. There is also one additional state denoted by $f$ (recall that the number of states in the tree is $\leqslant S - 1$).

We consider the same action set across each state: $\mathcal{A} = \big\{a_1, ..., a_{A-1}, a_f\big\}$. In the states of the binary tree where we have only described two actions, we can consider the other actions to remain in the same state deterministically with 0 cost.

The transitions and costs are defined as follows:

- For all states and actions in the tree that are not leaves, the transitions are specified above. The costs are all 0.

- For $s_i \in \mathcal{S}_\ell$: , and $a_j \in \mathcal{A}$

- if $j = f$, then $p(f|s_i, a_f) = 1$ and $c_k(s_i, a_f) = 0$.

- if $j \in \{1, ..., A-1\}$ and $j + (A-1) \cdot (i-1) \leqslant L$, then $p(g|s_i, a_j) = \frac{1}{T'}, p(s_i|s_i, a_j) = 1 - \frac{1}{T'}$ and the cost is an independent sample from a Bernoulli distribution at each episode $k$: $c_k(s_i, a_j) \sim \text{Ber}\left(\frac{D'}{2T'}\right)$.

- if $j \in \{1, ..., A-1\}$ and $j + (A-1) \cdot (i-1) > L$, then $a_j$ is the same as $a_f$, i.e. $p(f|s_i, a_j) = 1$ and $c_k(s_i, a_j) = 0$.

- For $f$,

  - $p(g|f, a_f) = \frac{1}{D'}, p(f|f, a_f) = 1 - \frac{1}{D'}$ and $c_k(f, a_f) = 1$.

  - for all $a_j \in \mathcal{A}\backslash\{a_f\}, p(f|f, a_j) = 1$ and $c_k(f, a_j) = 0$.

Denote the above distribution for $c_k$ by $\mathcal{D}$. In each episode there are at most $L + 1 \leqslant M$ non-zero costs, ensuring the condition on sparsity is respected.

For $i \in \{1, ..., N\}$, let $\mathcal{A}_i$ correspond to the actions in state $s_i \in \mathcal{S}_L$ which can transition directly to $g$ and $\mathcal{A}\backslash\mathcal{A}_i$ corresponds to the actions which deterministically transition to $f$ (e.g. if $(A-1) \cdot i \leqslant L$, then $\mathcal{A}_i = \{a_1, ..., a_{A-1}\}$). For any proper policy $\pi$ independent of the stochastically generated costs in episode $k$, we have

$$\mathbb{E}_{c_k \sim \mathcal{D}}\left[J_k^\pi(s_i)\right] = \mathbb{E}_{c_k \sim \mathcal{D}}\left[\sum_{a \in \mathcal{A}_i} \pi(a|s_i)\left(c_k(s_i, a) + \left(1 - \frac{1}{T'}\right)J_k^\pi(s_i)\right) + J_k^\pi(f)\sum_{a \notin \mathcal{A}_i} \pi(a|s_i)\right]$$

$$= \sum_{a \in \mathcal{A}_i} \pi(a|s_i)\left(\mathbb{E}_{c_k \sim \mathcal{D}}\left[c_k(s_i, a)\right] + \left(1 - \frac{1}{T'}\right)\mathbb{E}_{c_k \sim \mathcal{D}}\left[J_k^\pi(s_i)\right]\right) + D' \cdot \sum_{a \notin \mathcal{A}_i}^{A} \pi(a|s_i)$$

$$= \sum_{a \in \mathcal{A}_i} \pi(a|s_i)\left(\frac{D'}{2T'} + \left(1 - \frac{1}{T'}\right)\mathbb{E}_{c_k \sim \mathcal{D}}\left[J_k^\pi(s_i)\right]\right) + D' \cdot \left(1 - \sum_{a \in \mathcal{A}_i} \pi(a|s_i)\right)$$

$$\implies \mathbb{E}_{c_k \sim \mathcal{D}}\left[J_k^\pi(s_i)\right]\left(1 - \left(1 - \frac{1}{T'}\right)\sum_{a \in \mathcal{A}_i} \pi(a|s_i)\right) = \frac{D'}{2T'}\sum_{a \in \mathcal{A}_i} \pi(a|s_i) + D' \cdot \left(1 - \sum_{a \in \mathcal{A}_i} \pi(a|s_i)\right)$$

$$\implies \mathbb{E}_{c_k \sim \mathcal{D}}\left[J_k^\pi(s_i)\right] = \frac{\frac{D'}{2T'}\sum_{a \in \mathcal{A}_i} \pi(a|s_i) + D' \cdot \left(1 - \sum_{a \in \mathcal{A}_i} \pi(a|s_i)\right)}{1 - \left(1 - \frac{1}{T'}\right)\sum_{a \in \mathcal{A}_i} \pi(a|s_i)}$$

$$= \frac{D'}{2} \cdot \frac{\frac{1}{T'}\sum_{a \in \mathcal{A}_i} \pi(a|s_i) + 2\left(1 - \sum_{a \in \mathcal{A}_i} \pi(a|s_i)\right)}{\frac{1}{T'}\sum_{a \in \mathcal{A}_i} \pi(a|s_i) + \left(1 - \sum_{a \in \mathcal{A}_i} \pi(a|s_i)\right)}$$

$$\geqslant \frac{D'}{2}.$$

The optimal policy $\pi^\star$ is the policy that takes actions in the binary tree to reach state $s_{i^\star}$ and then $\pi^\star(a_{j^\star}|s_{i^\star}) = 1$ for $i^\star, j^\star = \arg\min_{i,j:j+(A-1)\cdot i \leqslant L} \sum_{k=1}^{K} c_k(s_i, a_j)$. We have $J_k^{\pi^\star}(s_0) = J_k^{\pi^\star}(s_{i^\star})$ and for any $k \geqslant 1$

$$J_k^{\pi^\star}(s_{i^\star}) = c_k(s_{i^\star}, a_{j^\star}) + \left(1 - \frac{1}{T'}\right)J_k^{\pi^\star}(s_{i^\star})$$

$$\implies J_k^{\pi^\star}(s_0) = T'c_k(s_{i^\star}, a_{j^\star})$$

$$\implies \sum_{k=1}^{K} J_k^{\pi^\star}(s_0) = T'\sum_{k=1}^{K} c_k(s_{i^\star}, a_{j^\star}) = T'\min_{i,j:j+(A-1)\cdot i \leqslant L}\sum_{k=1}^{K} c_k(s_i, a_j).$$

Hence,

$$\mathbb{E}_{c_1,...,c_K \overset{\text{iid}}{\sim} \mathcal{D}}\left[R_K\right] \geqslant \frac{D'}{2} \cdot K - T' \cdot \mathbb{E}_{c_1,...,c_K \overset{\text{iid}}{\sim} \mathcal{D}}\left[\min_{i,j:j+(A-1)\cdot i \leqslant L}\sum_{k=1}^{K} c_k(s_i, a_j)\right]$$

$$= T' \cdot \left( \frac{D'}{2T'} \cdot K - \mathbb{E}_{c_1, \dots, c_K \overset{\text{iid}}{\sim} \mathcal{D}} \left[ \min_{i,j : j+(A-1) \cdot i \leqslant L} \sum_{k=1}^{K} c_k(s_i, a_j) \right] \right)$$

$$= T' \cdot \mathbb{E}_{c_1, \dots, c_K \overset{\text{iid}}{\sim} \mathcal{D}} \left[ \max_{i,j : j+(A-1) \cdot i \leqslant L} \sum_{k=1}^{K} \left( \frac{D'}{2T'} - c_k(s_i, a_j) \right) \right]$$

We now apply Theorem G.1 with $p = 1 - \frac{D'}{2T'} \geqslant \frac{1}{2}$, $d = L \geqslant 100$ (since $S(A-1) \geqslant 400$ and $M \geqslant 101$) and $n = K \geqslant \frac{800 T_\star}{D} \log M \geqslant \frac{400 T'}{D'} \log M \geqslant 200 \frac{p}{1-p} \log d$. We obtain:

$$\sup_{c_1, \dots, c_K} \mathbb{E}\left[ R_K \right] \geqslant \mathbb{E}_{c_1, \dots, c_K \overset{\text{iid}}{\sim} \mathcal{D}} \left[ R_K \right] \geqslant 0.02 T' \sqrt{K \left( 1 - \frac{D'}{2T'} \right) \cdot \frac{D'}{2T'} \cdot \log L} - 1.5 T'$$

$$= \Omega\left( \sqrt{K T_\star D \log M} \right),$$

since $L \geqslant M/8$. Note that since $T_\star \geqslant D$, the hitting-time of the fast-policy is $D' + B + 2 = D$ and the hitting time of the optimal is $T' + B + 1 = T_\star$, as required. This concludes the proof. $\qquad\square$

## F  Lower bound under unknown transitions

**Theorem 5.1.** *For any $D, K, S, A$ with $S \geqslant 2, A \geqslant 16$, $D \geqslant 2$ and $K \geqslant SA$, there exists an SSP instance with $M = 1$, $S$ states and $A$ actions such that its diameter is $D$ and the expected regret for any learner without knowledge of the transitions after $K$ episodes is $\mathbb{E}[R_K] \geqslant \Omega\big(D\sqrt{SAK}\big)$.*

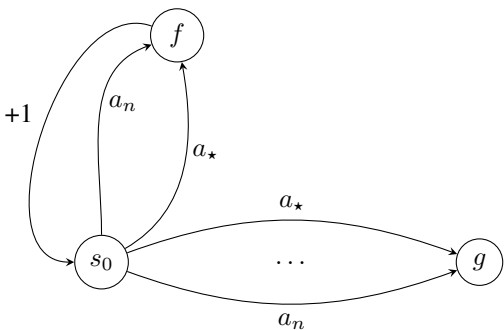

Figure 5: base case

*Proof.* The idea is to inject sparsity into the lower bound construction of [29] and to see if sparsity helps. Imagine the simple SSP in Figure 5, where at state $s_0$ there are $A$ available actions, all with zero cost, while in the state $f$ there is only one deterministic action with unit cost going back to $s_0$. Among them, there exists an action $a_\star$ such that the transition probabilities are given by: $P(g \mid s_0, a) = \frac{1}{D} - \epsilon \mathbb{I}(a \neq a_\star)$, and consequently, $P(f \mid s_0, a) = 1 - \frac{1}{D} + \epsilon \mathbb{I}(a \neq a_\star)$. The cost is therefore only suffered when the selected action transitions to the $f$ state. This will therefore not increase the hitting time of any proper deterministic policy while still inducing the desired sparsity.

Clearly, the optimal policy plays $a_\star$ at every time step to reach the goal as fast as possible and therefore $J^{\pi_\star}(s_0) = D - 1$.

Now, denote with $N_k$ the number of steps that the learner spends in $s_0$ in episode $k$ and $N_k^\star$ the number of steps that the learner picks action $a_\star$ in episode $k$. Note that $N_k$ is also the total cost that the learner suffers during episode $k$ minus one (since the last transition will not be paid). Thanks to our construction we can still prove Lemma C.1 in [29] as follows:

**Lemma F.1.**
$$\mathbb{E}\left[N_k\right] - 1 - J^{\pi_\star} = \epsilon \mathbb{E}\left[N_k - N_k^\star\right]$$

*Proof.*

$$\mathbb{E}\left[N_k\right] = \sum_{t=1}^{\infty} P[s_t = s_0]$$

$$= 1 + \sum_{t=2}^{\infty} P[s_t = s_0]$$

$$= 1 + \sum_{t=2}^{\infty} P[s_t = s_0 \mid s_{t+1} = s_0, a_{t-1} = a_\star]P[s_{t+1} = s_0, a_{t-1} = a_\star]$$

$$+ \sum_{t=2}^{\infty} P[s_t = s_0 \mid s_{t+1} = s_0, a_{t-1} \neq a_\star]P[s_{t+1} = s_0, a_{t-1} \neq a_\star]$$

$$= 1 + \left(1 - \frac{1}{D}\right)\sum_{t=2}^{\infty} P[s_{t+1} = s_0, a_{t-1} = a_\star] + \left(1 - \frac{1 - \epsilon}{D}\right)\sum_{t=2}^{\infty} P[s_{t+1} = s_0, a_{t-1} \neq a_\star]$$

Rearranging gives:

$$\mathbb{E}\left[N_k\right] - D = \epsilon \mathbb{E}\left[N_k - N_k^\star\right]$$

Adding and subtracting 1 gives the desired result. $\qquad\square$

Hence:

$$\mathbb{E}\left[R_K\right] = \sum_{k=1}^{K}\sum_{i=1}^{I_k} c_k(s_k^i, a_k^i) - \sum_{k=1}^{K} J^{\pi^\star}(s_0) = \mathbb{E}\left[N_k\right] - 1 - J^{\pi^\star} = \epsilon[N - N^\star]$$

where $N = \sum_{k=1}^{K} N_k$ and $N^\star = \sum_{k=1}^{K} N_k^\star$. Since we recovered Lemma C.1 in [29] as the starting block of the proof, following the derivation we can lower bound $N$ in expectation and upper bound the expected value of $N^\star$ to retrieve

**Lemma F.2** (Theorem C.4 in [29]). *Suppose that $D \geqslant 2$, $\epsilon \in (0, 1/8)$ and $A > 16$, for the problem described above we have:*

$$\mathbb{E}\left[R_K\right] \geqslant \epsilon K D \left(\frac{1}{8} - 2\epsilon\sqrt{\frac{2K}{A}}\right)$$

Now consider the following MDP. Let $\mathcal{S}$ be the set of states disregarding $g$ and $s_0$. The initial state $s_0$ has only one action which leads uniformly at random into one of the states $s \in \mathcal{S}$, where each one has its own optimal action $a_s^\star$. Then the transition distributions are defined $P(g \mid a_s^\star, s) = 1/D, P(s \mid a_s^\star, s) = 1 - 1/D$, and $P(g \mid a, s) = (1 - \epsilon)/D$, $P(s \mid a, s) = 1 - (1 - \epsilon)/D$ for any other action $a \in \mathcal{A}\backslash\{a_s^\star\}$. Note that for each state, the learner is faced with a simple problem as the one described above. Therefore, we can apply Lemma F.2 for each state separately and lower bound the learner's expected regret the sum of the regrets suffered at each state, which would depend on the number of times each state $s \in \mathcal{S}$ is visited from the initial state. Since reaching each state has uniform probability, there are many states (constant fraction) that are chosen $\Theta(K/S)$ times. Summing the regret bounds and choosing $\epsilon$, gives the desired bound.

Denote by $K_s$ the number of episodes the state $s \in \mathcal{S}$ is visited:

$$\mathbb{E}\left[R_K\right] \geqslant \sum_{s\in\mathcal{S}} \mathbb{E}\left[\epsilon K_s D\left(\frac{1}{8} - 2\epsilon\sqrt{\frac{2K_s}{A}}\right)\right] = \frac{\epsilon K D}{8} - 2\epsilon^2 D\sqrt{\frac{2}{A}}\sum_{s\in\mathcal{S}}\mathbb{E}\left[K_s^{3/2}\right]$$

Then:

$$\sum_{s\in\mathcal{S}}\mathbb{E}\left[K_s^{3/2}\right] \leqslant \sum_{s\in\mathcal{S}}\sqrt{\mathbb{E}\left[K_s\right]}\sqrt{\mathbb{E}\left[K_s^2\right]} = \sum_{s\in\mathcal{S}}\sqrt{\mathbb{E}\left[K_s\right]}\sqrt{\mathbb{E}\left[K_s^2\right] + \mathbb{V}\left[K_s\right]} = \sum_{s\in\mathcal{S}}\sqrt{\frac{K}{S}}\sqrt{\frac{K^2}{S^2} + \frac{K(S-1)}{S^2}} \leqslant K\sqrt{\frac{2K}{S}}$$

Leading to:

$$\mathbb{E}\left[R_K\right] \geqslant \frac{\epsilon K D}{8} - 2\epsilon^2 D K\sqrt{\frac{2K}{SA}} \geqslant \frac{1}{1024}D\sqrt{SAK}$$

for $\epsilon = 1/64\sqrt{SA/K}$ $K \geqslant SA$, concluding the proof. $\qquad\square$

# G   Lower bound on the maximum of asymmetric zero-mean random walks

We extend the lower bound of [25] to asymmetric zero-mean random-walks. We consider $p \geqslant 1/2$ because it simplifies the proof below (lower-bounding $\psi$ by 1 and upper-bounding $C$ in proof below) and is what we need in the proof of Section 4.6 in Appendix E (we use $p = 1 - D/2T^\star$).

**Theorem G.1.** *Fix $p \in [\frac{1}{2}, 1 - \frac{1}{n}]$. Consider random walks $Z_i^{(n)} = \sum_{t=1}^n X_t^i$, where*

$$X_t^i = \begin{cases} -p, & \text{w.p. } 1 - p \\ 1 - p, & \text{w.p. } p. \end{cases}$$

*If $n \geqslant 200 \frac{p}{1-p} \log d$ (also ensures that $p \leqslant 1 - \frac{1}{n}$) and $d \geqslant 100$. Then,*

$$\mathbb{E}\big[\max_{1 \leqslant i \leqslant d} Z_i^n\big] \geqslant 0.02\sqrt{np(1-p)\log d} - 1.5.$$

*Proof.* We follow the same lines as [25] who show a special case of the result for $p = 1/2$. We generalize it to $p > 1/2$.

Consider $Z^{(n)} = \sum_{t=1}^n X_t$, a random-walk of length $n$, then $B_n = Z^{(n)} + pn \sim B(n, p)$, Binomial distribution with parameters $n$ and $p$.

## G.1   1st part of the proof:

The 1st part of the proof is all about providing a lower bound on $\mathbb{P}\big(B_n \geqslant pn + t - 1\big)$ in (10) for any $t \in [1, np + 1]$.

**Lemma G.2** (Generalized version of Lemma 4 of [25], Theorem 2 of [21])**.** *Let $n, k$ be integers satisfying $n \geqslant 1$ and $pn \leqslant k \leqslant n$. Define $x = \frac{k - pn}{\sqrt{p(1-p)n}}$. Then $B_n \sim B(n, p)$ satisfies*

$$\mathbb{P}\big(B_n \geqslant k\big) \geqslant \sqrt{n}\binom{n-1}{k-1}p^{k-1/2}(1-p)^{n-k+1/2} \cdot \frac{1 - \Phi(x)}{\phi(x)},$$

*where $\phi(x)$ is the PDF of a standard Normal and $\Phi(x)$ is the CDF. The proof can be found in [21].*

Denote $D(p, q) = p \log \frac{p}{q} + (1 - p) \log \frac{1-p}{1-q}$ as the KL-divergence between two Bernoullis.

**Lemma G.3** (Generalized version of Theorem 5 of [25])**.** *Let $n, k$ be integers satisfying $n \geqslant 1$, and $np \leqslant k \leqslant n$. Define $x = \frac{k - pn}{\sqrt{p(1-p)n}}$. Then $B_n \sim B(n, p)$ satisfies*

$$\mathbb{P}\big(B_n \geqslant k\big) \geqslant \frac{\exp\big(-nD\big(\frac{k}{n}, p\big)\big)}{e^{1/6}\sqrt{2\pi}} \cdot \frac{1 - \Phi(x)}{\phi(x)}.$$

*Proof.* For $k = n$, we verify the statement of the theorem directly. The left hand side is $\mathbb{P}\big(B_n \geqslant n\big) = p^n$. The right hand side is smaller because $\exp\big\{-nD\big(1, p\big)\big\} = p^n$ and for $x = \sqrt{n\frac{1-p}{p}} > 0$, we have $\frac{1-\Phi(x)}{\phi(x)} \leqslant \sqrt{2}$ (see e.g. Section 3.3 in [15]).

For $np \leqslant k < n$, we first bound the binomial coefficient $\binom{n}{k}$. Stirling's formula for the factorial [27] gives for any $n \geqslant 1$,

$$\sqrt{2\pi n}\Big(\frac{n}{e}\Big)^n < n! < e^{1/12}\sqrt{2\pi n}\Big(\frac{n}{e}\Big)^n.$$

Since $0 < np \leqslant k \leqslant n - 1$, we can use this approximation for $k, n$ and $n - k$ and obtain

$$\binom{n}{k} = \frac{n!}{k!(n-k)!}$$

$$> \frac{n^n e^{-n}\sqrt{2\pi n}}{(e^{1/12}k^k e^{-k}\sqrt{2\pi k}) \cdot (e^{1/12}(n-k)^{n-k}e^{-(n-k)}\sqrt{2\pi(n-k)})}$$

$$= \frac{1}{e^{1/6}\sqrt{2\pi}}\Big(\frac{n}{k}\Big)^k\Big(\frac{n}{n-k}\Big)^{n-k}\sqrt{\frac{n}{k(n-k)}}$$

$$= \frac{1}{e^{1/6}\sqrt{2\pi}} \frac{1}{p^k(1-p)^{n-k}} \exp\left\{-nD\left(\frac{k}{n},p\right)\right\}\sqrt{\frac{n}{k(n-k)}},$$

since

$$D\left(\frac{k}{n},p\right) = \frac{k}{n}\log\left(\frac{k}{np}\right) + \left(1 - \frac{k}{n}\right)\log\left(\frac{1-k/n}{1-p}\right) = -\frac{k}{n}\log\left(\frac{np}{k}\right) - \frac{n-k}{n}\log\left(\frac{n(1-p)}{n-k}\right)$$

$$\implies \exp\left\{-nD\left(\frac{k}{n},p\right)\right\} = \left(\frac{np}{k}\right)^k \cdot \left(\frac{n(1-p)}{n-k}\right)^{n-k} = p^k(1-p)^{n-k}\left(\frac{n}{k}\right)^k\left(\frac{n}{n-k}\right)^{n-k}.$$

Since $k \geqslant 1$, we can write the binomial coefficient as $\binom{n-1}{k-1} = \frac{k}{n}\binom{n}{k}$. By Lemma G.2, we have

$$\mathbb{P}\big(B_n \geqslant k\big) \geqslant \sqrt{n}\binom{n-1}{k-1}p^{k-1/2}(1-p)^{n-k+1/2} \cdot \frac{1-\Phi(x)}{\phi(x)}$$

$$= \sqrt{n}\frac{k}{n}\binom{n}{k}p^{k-1/2}(1-p)^{n-k+1/2} \cdot \frac{1-\Phi(x)}{\phi(x)}$$

$$\geqslant \frac{1}{e^{1/6}\sqrt{2\pi}}\frac{k}{\sqrt{n}}\sqrt{\frac{n}{k(n-k)}}\frac{p^{k-1/2}(1-p)^{n-k+1/2}}{p^k(1-p)^{n-k}}\exp\left\{-nD\left(\frac{k}{n},p\right)\right\} \cdot \frac{1-\Phi(x)}{\phi(x)}$$

$$= \frac{1}{e^{1/6}\sqrt{2\pi}}\sqrt{\frac{k}{n-k}} \cdot \sqrt{\frac{1-p}{p}}\exp\left\{-nD\left(\frac{k}{n},p\right)\right\} \cdot \frac{1-\Phi(x)}{\phi(x)}.$$

The result follows from $\sqrt{\frac{k}{n-k}} \geqslant \sqrt{\frac{np}{n-np}} = \sqrt{\frac{p}{1-p}}$ for $np \leqslant k \leqslant n-1$. $\qquad\square$

For $k = pn + xn$, the 2nd-order Taylor approximation of $u(x) = D\left(\frac{k}{n},p\right) = D(p+x,p)$ around $0$ is $\frac{x^2}{2p(1-p)}$. We define $\psi : \left[-p, 1-p\right] \to \mathbb{R}$ as the ratio of the divergence and the approximation:

$$\psi(x) = D(p+x,p) \cdot \frac{2p(1-p)}{x^2}.$$

In particular, we have that $1 \leqslant \psi(x) \leqslant \frac{p(1-p)}{(x+p)(1-p-x)}$ for $x \in [0, 1-p]$. This can be shown using Taylor's theorem on $u(x)$: for some $z \in [0, x]$,

$$D(p+x,p) = \frac{u^{(2)}(z)}{2}x^2 = \left(\frac{1}{z+p} + \frac{1}{1-p-z}\right)\frac{x^2}{2} = \frac{x^2}{2(z+p)(1-p-z)}$$

$$\implies \frac{x^2}{2p(1-p)} \leqslant D(p+x,x) \leqslant \frac{x^2}{2(x+p)(1-p-x)}, \tag{9}$$

since $\frac{1}{(x+p)(1-p-x)}$ is increasing on $[0, 1-p)$.

Let $t \in [1, np+1]$ be a real number. By Lemma G.3 and Lemma 1 in [25] (also Mill's ratio for standard Gaussian [2]), we have

$$\mathbb{P}\big(B_n \geqslant pn + t - 1\big) = \mathbb{P}\big(B_n \geqslant \lceil pn + t - 1\rceil\big)$$

$$\geqslant \frac{\exp\left(-nD\left(\frac{\lceil pn+t-1\rceil}{n},p\right)\right)}{e^{1/6}\sqrt{2\pi}} \cdot \frac{\pi}{\pi\frac{\lceil pn+t-1\rceil-np}{\sqrt{p(1-p)n}} + \sqrt{2\pi}}$$

$$\geqslant \frac{\exp\left(-nD\left(\frac{pn+t}{n},p\right)\right)}{e^{1/6}\sqrt{2\pi}} \cdot \frac{\pi}{\pi\frac{pn+t-np}{\sqrt{p(1-p)n}} + \sqrt{2\pi}}$$

$$= \frac{\exp\left(-nD\left(p + \frac{t}{n},p\right)\right)}{e^{1/6}\sqrt{2\pi}} \cdot \frac{\pi}{\pi\frac{t}{\sqrt{p(1-p)n}} + \sqrt{2\pi}}$$

$$= e^{-1/6}\exp\left(-\frac{1}{2p(1-p)}\psi\left(\frac{t}{n}\right) \cdot \frac{t^2}{n}\right) \cdot \frac{1}{\frac{\sqrt{2\pi}t}{\sqrt{np(1-p)}} + 2}. \tag{10}$$

## G.2   2nd part of the proof:

We can now turn to the actual proof of the result. Define the event $A$ equal to the case that at least one of the $Z_i^{(n)}$ is greater or equal to $C\sqrt{np(1-p)\log d}-1$. We will show this event / threshold controls the expectation of the maximum. First, we define $C$ and provide some upper and lower bounds for it. Denote by $f(d)=\sqrt{2-\frac{2\log\log d}{\log d}}$, then

$$C=C(d,n)=\frac{1}{\sqrt{\psi\left(\sqrt{\frac{2p(1-p)\log d}{n}}\right)}}\sqrt{2-\frac{2\log\log d}{\log d}}=\frac{1}{\sqrt{\psi\left(\sqrt{\frac{2p(1-p)\log d}{n}}\right)}}f(d). \quad (11)$$

We bound the two factors separately:

- $z=\sqrt{\frac{2p(1-p)\log d}{n}}\in[0,\frac{1}{10}(1-p)]$ for $n\geqslant 200\frac{p}{1-p}\log d$ and so

$$1\leqslant\psi(z)\leqslant\frac{p(1-p)}{(z+p)(1-p-z)}\leqslant\frac{1-p}{(1-p-\frac{1-p}{10})}=\frac{10}{9}. \quad (12)$$

- The function $f(d)$ is as in [25]: decreasing on $(1,e^e]$, increasing on $[e^e,+\infty)$, and $\lim_{d\to\infty}f(d)=\sqrt{2}$. Therefore for all $d\in[5,\infty)$,

$$1.12\leqslant f(e^e)\leqslant f(d)\leqslant\max\{f(5),\sqrt{2}\}=\sqrt{2}$$

This gives for $n\geqslant 200\frac{p}{1-p}\log d$,

$$1\leqslant\frac{1.12}{\sqrt{10/9}}\leqslant C(d,n)\leqslant\sqrt{2} \quad (13)$$

Since $p\geqslant 1/2$, if $n\geqslant 200\frac{p}{1-p}\log d$, then $n>\frac{200}{p(1-p)\log d}$ (if $d\geqslant 8$) and $n\geqslant 200\frac{1-p}{p}\log d$. The above implies:

$$1<C\sqrt{np(1-p)\log d}\leqslant np\leqslant np+1. \quad (14)$$

Finally, we bound the quantity of interest:

$$\mathbb{E}\left[\max_{1\leqslant i\leqslant d}Z_i^n\right]=\mathbb{E}\left[\max_{1\leqslant i\leqslant d}Z_i^n|A\right]\cdot\mathbb{P}(A)+\mathbb{E}\left[\max_{1\leqslant i\leqslant d}Z_i^n|A^C\right]\cdot\left(1-\mathbb{P}(A)\right)$$

$$\geqslant\mathbb{E}\left[\max_{1\leqslant i\leqslant d}Z_i^n|A\right]\cdot\mathbb{P}(A)+\mathbb{E}\left[Z_1^{(n)}|A^C\right]\cdot\left(1-\mathbb{P}(A)\right)$$

$$=\mathbb{E}\left[\max_{1\leqslant i\leqslant d}Z_i^n|A\right]\cdot\mathbb{P}(A)+\mathbb{E}\left[Z_1^{(n)}|Z_1^{(n)}<C\sqrt{np(1-p)\log d}-1\right]\cdot\left(1-\mathbb{P}(A)\right)$$

$$\geqslant\mathbb{E}\left[\max_{1\leqslant i\leqslant d}Z_i^n|A\right]\cdot\mathbb{P}(A)+\mathbb{E}\left[Z_1^{(n)}|Z_1^{(n)}\leqslant 0\right]\cdot\left(1-\mathbb{P}(A)\right)\quad\text{by (14)}$$

$$\geqslant\left(C\sqrt{np(1-p)\log d}-1\right)\cdot\mathbb{P}(A)+\mathbb{E}\left[Z_1^{(n)}|Z_1^{(n)}\leqslant 0\right]\cdot\left(1-\mathbb{P}(A)\right). \quad (15)$$

**First, we lower bound** $\mathbb{E}\left[Z_1^{(n)}|Z_1^{(n)}\leqslant 0\right]$. Let $\beta=\frac{1}{1-\sqrt{\frac{2n}{\pi[n(1-p)][(n-\lceil n(1-p)\rceil)]}}}$. For $n\geqslant\frac{200}{p(1-p)}\log d$,

we have $n\geqslant\frac{205}{\pi p(1-p)}\geqslant\frac{200+\pi p}{\pi(1-p)p}$ and $\beta\leqslant\frac{10}{9}$.

Then Lemma 2.2 in [26] combined with Lemma 8 in [13] give that for $Y_n\sim B(n,1-p)$:

$$\mathbb{E}\left[Y_n|Y_n\geqslant n(1-p)\right]<n(1-p)+\beta\sqrt{np(1-p)}<n(1-p)+\frac{10}{9}\sqrt{np(1-p)}.$$

Since $B_n=Z^{(n)}+pn\sim B(n,p)$ can be written as $n-Y_n$, we have:

$$\mathbb{E}\left[Z_1^{(n)}|Z_1^{(n)}\leqslant 0\right]=\mathbb{E}\left[B_n|B_n\leqslant np\right]-np$$

$$= \mathbb{E}\Big[n - Y_n | n - Y_n \leqslant np\Big] - np$$

$$= n - \mathbb{E}\Big[Y_n | Y_n \geqslant n(1-p)\Big] - np$$

$$\geqslant n - n(1-p) - \frac{10}{9}\sqrt{np(1-p)} - np$$

$$= -\frac{10}{9}\sqrt{np(1-p)}. \tag{16}$$

**Next, we lower-bound $\mathbb{P}(A)$:**

$$\mathbb{P}(A) = 1 - \mathbb{P}(A^C)$$

$$= 1 - \left(\mathbb{P}\Big[Z_1^{(n)} < C\sqrt{np(1-p)\log d} - 1\Big]\right)^d$$

$$= 1 - \left(\mathbb{P}\Big[B_n < np + C\sqrt{np(1-p)\log d} - 1\Big]\right)^d$$

$$= 1 - \left(1 - \mathbb{P}\Big[B_n \geqslant np + C\sqrt{np(1-p)\log d} - 1\Big]\right)^d$$

$$\geqslant 1 - \exp\left(-d \cdot \mathbb{P}\Big[B_n \geqslant np + C\sqrt{np(1-p)\log d} - 1\Big]\right) \quad \text{since } 1 - x \leqslant e^{-x}$$

$$\geqslant 1 - \exp\left(-d \cdot \frac{e^{-1/6}\exp\left(-\frac{1}{2p(1-p)}\psi\big(\frac{C\sqrt{np(1-p)\log d}}{n}\big) \cdot \frac{C^2 np(1-p)\log d}{n}\right)}{\frac{\sqrt{2\pi}C\sqrt{np(1-p)\log d}}{\sqrt{np(1-p)}} + 2}\right) \quad \text{using (10) and (14)}$$

$$= 1 - \exp\left(-d \cdot \frac{e^{-1/6}\exp\left(-\frac{1}{2}\psi\big(C\sqrt{p(1-p)\log d/n}\big) \cdot C^2 \log d\right)}{\sqrt{2\pi}C\sqrt{\log d} + 2}\right)$$

$$= 1 - \exp\left(-\frac{e^{-1/6}d^{1-\frac{C^2}{2}\psi\big(C\sqrt{p(1-p)\log d/n}\big)}}{\sqrt{2\pi}C\sqrt{\log d} + 2}\right)$$

$$\geqslant 1 - \exp\left(-\frac{e^{-1/6}d^{1-\frac{C^2}{2}\psi\big(\sqrt{\frac{2p(1-p)\log d}{n}}\big)}}{2\sqrt{\pi\log d} + 2}\right) \quad \text{by (13)}.$$

We now use that $d^{1-\frac{C^2}{2}\psi\big(\sqrt{2p(1-p)\log d/n}\big)} = \log d$ by the definition of $C$ in (11). Hence, we obtain:

$$\mathbb{P}(A) \geqslant 1 - \exp\left(-\frac{e^{-1/6}\log d}{2\sqrt{\pi\log d} + 2}\right) = 1 - g(d), \quad \text{for } g(d) = \exp\left(-\frac{e^{-1/6}\log d}{2\sqrt{\pi\log d} + 2}\right). \tag{17}$$

**Putting everything together:** we plug (16) and (17) into (15) to get

$$\mathbb{E}\Big[\max_{1\leqslant i\leqslant d} Z_i^n\Big] \geqslant \big(C\sqrt{np(1-p)\log d} - 1\big) \cdot \big(1 - g(d)\big) - \frac{10}{9}g(d)\sqrt{np(1-p)}$$

$$= \frac{f(d) \cdot (1 - g(d))}{\sqrt{\psi\big(\sqrt{\frac{2p(1-p)\log d}{n}}\big)}}\big(\sqrt{np(1-p)\log d} - 2\big) - \frac{10}{9}g(d)\sqrt{np(1-p)} \quad \text{using (11)}$$

$$\geqslant \frac{f(d)\big(1 - g(d)\big)}{\sqrt{10/9}}\big(\sqrt{np(1-p)\log d} - 2\big) - \frac{10}{9}g(d)\sqrt{np(1-p)} \quad \text{using (12)}$$

$$= \sqrt{np(1-p)\log d} \cdot \left(\frac{f(d)\big(1 - g(d)\big)}{\sqrt{10/9}} - \frac{10}{9} \cdot \frac{g(d)}{\sqrt{\log(d)}}\right) - 2\frac{f(d)\big(1 - g(d)\big)}{\sqrt{10/9}}$$

$$\geqslant 0.02\sqrt{np(1-p)\log d} - 1.5,$$

for $d \geqslant 100$ (we also used that $\sqrt{np(1-p)\log d} > 2$ in the 2nd inequality). This gives the result. $\qquad \square$

