# OpenReview forum: "Stochastic Shortest Path with Sparse Adversarial Costs"
_NeurIPS.cc/2025/Conference — NeurIPS 2025 poster_

### Official Review · Reviewer_fyrC · 2025-06-18

**Clarity:** 4
**Significance:** 3
**Originality:** 3
**Rating:** 5
**Confidence:** 3

**Summary:**

The paper addresses adversarial Stochastic Shortest Pash (SSP) with sparse costs under full-information feedback. The main take is that such sparsity should lead to a regret bound scaling with some measure of sparsity of the problem, and not with the space size. Indeed, while negative entropy regularised OMD leads to bounds scaling as SA in general, the authors show that it scales as S on sparse problem. Motivated by this, they propose a new regularisation for OMD (smoothly interpolating between neg-entropy and l2 norm) which leads to a regret scaling with the sparsity of the problem. Additionally, the authors instantiate a parameter-free version of the same algorithm that does not require knowledge of the sparsity level nor the hitting time of the optimal policy. Finally, the authors show lower bounds ( for some regimes of the sparsity factor M) matching the upper bound over these instances. The authors also show that this bound does not hold under bandit feedback, in which the M factor is substituted by SA again, regardless of the sparsity, suggesting that without further assumptions, sparsity cannot be exploited effectually under non-full feedback settings.


**Overall Motivation for Rating**:

In general, I am not strongly leaning towards accept just because (1) I have *not* carefully read the proofs and (2) I have some knowledge of the related works but I would not call me an expert. My current score will change conditioned on: a satisfactory rebuttal showing that my questions are not so well founded; whether other reviewers raise some major concerns I was not aware of but I might agree with ex-post.

**Questions:**

- The lower bound goes towards non-sparse problem more than sparse ones, as it holds for M > 100. Do the authors have a take on this, or some intuition on what could happen for sparser problem (M<100)?
- Can the authors better articulate on the limitations suggested by Thm 5.1? Which sub-portions of the algorithmic solutions could be nonetheless employed for unknown transition settings?

My current score will be undergo the following changes:
- increased: whether my questions will be addressed properly, and further discussions about them will be included in the manuscript.
- decreased: whether other reviewers raise some *major* concerns I was not aware of but I might agree with post discussion.
- Kept the same: whether the answers were not addressed.

**Ethical Concerns:**

["NO or VERY MINOR ethics concerns only"]

**Final Justification:**

The paper addresses a relevant problem and introduces an interesting analyses, and as all my concerns were addressed in the rebuttal phase I am positive to raise my score from 4 to 5.
Nonetheless, I missed some relevant typos and factors noted by other reviewers, thus I might not be able to champion this paper with high confidence, I might need a discussion with other reviewers then.

**Limitations:**

Yes

**Quality:**

3

**Strengths And Weaknesses:**

In the following, I will refer to Strengths with (S) and Weaknesses with (W), line numbering with (l#)

**Quality**:

(S) The work reaches a good quality overall and it is self-sufficient in covering a problem and pointing to limitations and next step.

**Clarity**:

(S) The paper is extremely well written, it does introduce all the concepts clearly and it does address all the relevant limitations.

(Minor W) I would format equation under l170 over a single line.

**Significance**:

(Minor W) Full-information feedback might sound limiting, yet I believe that (S) a formal characterisation of the problem is still relevant and provides an important stepping stone towards more realistic settings, and the authors provided enough tools to go towards this direction.

**Originality**:

(S) The proposed regularisation is interesting and novel (as far as I am aware of), and the counterexamples on standard neg-entropy OMD limitations are a pretty nice rethorical tool.
(Very Minor W) Most of the other proof structures of the lower bounds are based on previous works, which is yet pretty common.

---

> ### Author Rebuttal · Authors · 2025-07-28
>
> We thank the reviewer for their thorough feedback. We address your comments and question below.
>
> # Weaknesses:
>
> 1. (Minor W) I would format equation under l170 over a single line.
>
> - **Thank you for this suggestion.**
>
> # Questions
>
> Q1: The lower bound goes towards non-sparse problem more than sparse ones, as it holds for M > 100. Do the authors have a take on this, or some intuition on what could happen for sparser problem (M<100)?
>
> - **We don’t believe that this condition on $M > 100$ is necessary. It appears from the technical result on the expectation of the maximum of $d$ asymmetric zero-mean random-walks (Theorem F.1). We prove this result for $d \geq 100$ but do not believe this condition is necessary. The bottleneck is really this specific technical result and provided a more refined analysis of this expectation becomes available, this would immediately apply to our SSP lower-bound. However, achieving this refined result is a highly technical non-trivial task.**
>
> Q2: Can the authors better articulate on the limitations suggested by Thm 5.1? Which sub-portions of the algorithmic solutions could be nonetheless employed for unknown transition settings?
>
> - **Theorem 5.1 highlights a key limitation in the unknown transitions setting: it establishes that sparsity in the MDP structure does not eliminate the $\sqrt{SA}$ dependence in the regret bound, even in sparse instances. This result is somewhat surprising, as it suggests that the learner cannot leverage sparsity to bypass the worst-case dependence on the size of the state-action space.**
>
> - **Nonetheless, this lower bound does not preclude the possibility that sparsity may still offer improvements within the unavoidable dependence, it could indeed still reduce the polynomial dependence though not completely remove it. In this sense, components of our algorithm, including our sparsity-aware regularizer, could still be beneficial in the unknown-transition setting, even if the full elimination of $\sqrt{SA}$ is not possible.**
>
> - **Our decision not to explore this setting comes also from the fact that even in the non-sparse case, the optimal regret bound is still an open question, with a polynomial gap of $\sqrt{S}$ between the best known upper and lower bound [5] . Without a minimax characterization of the optimal non-sparse regret, it is premature to rigorously quantify the gains brought by sparsity. We included Theorem 5.1 primarily to provide a conceptual insight: while sparsity helps in the main setting we considered, it does not change the fundamental lower bound structure in the unknown transition case, reinforcing the complexity of the problem.**
>
> - **Further work is needed to understand which algorithmic techniques remain effective under unknown transitions, and how sparsity can be exploited in this regime.**
>
>
> [5] Chen, Liyu, and Haipeng Luo. "Finding the stochastic shortest path with low regret: The adversarial cost and unknown transition case." International Conference on Machine Learning. PMLR, 2021.

---

> > ### Comment · Reviewer_fyrC · 2025-08-04
> >
> > Thank you for addressing my concerns, I believe these answers to be satisfactory. I am more than positive raising my score, even thought I believe a discussion with other reviewers might be needed anyway, due to their concerns as well.

---

### Official Review · Reviewer_Txzu · 2025-06-20

**Clarity:** 2
**Significance:** 3
**Originality:** 3
**Rating:** 4
**Confidence:** 3

**Summary:**

The authors study the stochastic shortest path problem under sparsity. In particular, they show the online mirror descent (OMB) with entropy regularization does not take advantage of sparsity in general, and introduce a new family of regularizers which can. With the new regularizers, they prove a bound on the regret of OMB which scales with the sparsity instead of the absolute size of the state space. They also prove a lower bound which matches their upper bound up to logarithmic factors.

**Questions:**

# Questions:
- In the definition of D (equation 148) you swap the min and max. Why is this justified?
- Theorem 3.1 is very interesting. Do you have any idea how common these instances are in which negative-entropy regularization fails to get the right (ie sparse) rate? Is it possible to generalize your proof to a statement such as: For any S, any M and any A, there exists an SSP instance with regret $\Omega(\sqrt{DKT_* \log (SAT_*)})$? That is, showing that for all parameters, there exists an SSP instance which won’t take advantage of the sparsity.
- Should $K\geq MT$ be $K\geq MT_*$ in the statement of theorem 4.3? Otherwise what is T?
- Couldn’t the bound of Theorem 4.3 be worse than the status quo if K is very large? The second term vanishes, but if K is large enough such that $\log(\log(K)\sqrt{T_\*}) > \log(T^\*SA)$  then theorem 4.3 is looser than $O(DKT_*\log(SAT^*))$? So perhaps you need to state the result with an upper bound on K as well?

# Minor comments
- Line 142: “a proper policy” -> “one proper policy”
- Maybe move line 158 upwards to where you are discussing hitting times? Feels out of place
- Why does $\eta$ matter in the statement of Theorem 4.1? It’s just a parameter in the proof no?
- Missing square root on the regret in line 273 I think

**Ethical Concerns:**

["NO or VERY MINOR ethics concerns only"]

**Final Justification:**

Maintaining my original score. I appreciate that the gains are not massive, but they are still gains.

**Limitations:**

yes

**Quality:**

3

**Strengths And Weaknesses:**

# Strengths
- Sparsity is a very natural setting to consider.
- They provide a new algorithm and both upper and lower bounds

# Weaknesses
- A bit hard to follow for someone who is not an expert in the area
- Some concerns I have about Theorem 4.3; see below.

---

> ### Author Rebuttal · Authors · 2025-07-28
>
> We thank the reviewer for their constructive and thorough feedback. We address your comments, suggestions and questions below.
>
> # Weaknesses
> 1. A bit hard to follow for someone who is not an expert in the area.
> - **We are aware that this setting of adversarial SSP problems is quite notation-heavy and contains a lot of technical concepts. We tried to present the setting as clearly as possible.**
>
> # Questions
>
> Q1: In the definition of D (equation 148) you swap the min and max. Why is this justified?
>
> - **Sorry for the confusing typo, as the reviewer points out the definition is actually max_{s}min_{\pi} the correct definition.**
>
> Q2: Theorem 3.1 is very interesting. Do you have any idea how common these instances are in which negative-entropy regularization fails to get the right (ie sparse) rate? Is it possible to generalize your proof to a statement such as: For any S, any M and any A, there exists an SSP instance with regret ? That is, showing that for all parameters, there exists an SSP instance which won’t take advantage of the sparsity.
>
> - **The SSP instance we constructed is specifically designed to make the negative-entropy regularization fail. While we tried other constructions on which it did not always fail, it is unclear how pervasive this behaviour is more generally. We believe it should be possible to extend this result beyond $M = 3$ and $A=2$  but at the expense of an even more tedious analysis so since the purpose of this result was to show that Theorem 4.1 (our sparse upper-bound) cannot be achieved with negative entropy regularization, we presented Theorem 3.1.**
>
> Q3: Should $K > MT$ be $K > MT_\star$ in the statement of theorem 4.3? Otherwise what is T?
>
> - **This is again a typo sorry, we will fix it in the camera ready.**
>
> Q4: Couldn’t the bound of Theorem 4.3 be worse than the status quo if K is very large? The second term vanishes, but if K is large enough such that  then theorem 4.3 is looser than ? So perhaps you need to state the result with an upper bound on K as well?
>
> - **Yes this correct though usually the parameter free results obtained with expert-like algorithms are consider up to loglog factors (e.g. [3, 4] for references), we will clarify better such a discussion in the camera ready version.**
>
> # Minor comments
>
> Why does $\eta$ matter in the statement of Theorem 4.1? It’s just a parameter in the proof no?
>
> - **Yes, it is how the eta is set in the algorithm in order to obtain the result, it is in the theorem statement for completeness.**
>
> - **Thank for you for the useful other minor comments and suggestions - we will amend these in the final version.**
>
> [3] Cutkosky, Ashok, and Francesco Orabona. "Black-box reductions for parameter-free online learning in banach spaces." Conference On Learning Theory. PMLR, 2018.
>
> [4] Jacobsen, Andrew, and Ashok Cutkosky. "Parameter-free mirror descent." Conference on Learning Theory. PMLR, 2022.

---

> > ### Comment · Reviewer_Txzu · 2025-08-09
> >
> > Thank you for answering my questions. Please make sure the typos are fixed if the paper is accepted. I'm maintaining my score.

---

### Official Review · Reviewer_9PaL · 2025-06-30

**Clarity:** 3
**Significance:** 2
**Originality:** 2
**Rating:** 3
**Confidence:** 4

**Summary:**

This work considers the adversarial stochastic shortest path (SSP) problem with sparse costs in the full-information feedback setting. With known transition P, this work improves the previous best known result $O(\sqrt{DK{T_{\*}} \log ({T_{\*}}SA)})$ to $O(\sqrt{DKT\_{\*} \log (T\_{\*}M)})$, where $D$ is the diameter, $K$ is the number of episodes, ${T_{\*}}$ is the expected hitting time of the optimal policy, $M$ is the sparsity parameter, and $S$ and $A$ are the state and action space size, respectively. This is achieved by a new regularizer proposed by the authors, which regularizes the (1+1/p)-norm of the occupancy measures. The authors propose two algorithms, one requiring the knowledge of $M$ and the other being a parameter-free one, without requiring the knowledge of $M$. For the same problem, the authors prove a lower bound of ${\Omega}(\sqrt{DK{T\_{\*}}\log (M)})$, showing the tightness of the dependence on $M$ in the regret upper bound. The authors also prove an algorithm-specific lower bound, showing that the canonical Shannon entropy is not able to achieve the same regret upper bound. With unknown transition P, the authors prove a regret lower bound $\Omega(D\sqrt{SAK})$.

**Questions:**

1. Could the authors please further explain the intuition on Line 223-229? I do not quite understand why “inducing an OMD update at points near the boundary of the space that can only move to new points in very close proximity” prevents the algorithms from obtaining a sparsity-dependent bound.

**Ethical Concerns:**

["NO or VERY MINOR ethics concerns only"]

**Final Justification:**

My concern regarding the significance of the results remains unresolved.

**Limitations:**

yes

**Quality:**

3

**Strengths And Weaknesses:**

**Strengths**
1. The results are new.

**Weaknesses**
1. **Significance**: Though the results are new to the literature, I am a bit concerned about the significance of the results. In particular, the authors obtain the upper bound of $O(\sqrt{DK{T_{\*}} \log (T_{\*}M)})$ in the full-information setting. Though the dependence on $M$ indeed matches the proved lower bound, the improvement from $O(\sqrt{\log (SA)})$ to $O(\sqrt{\log M})$ against previous work [1] does not seem to be very valuable to me. Further, I find that the improvement does not even recover it in the special case of the expert setting (i.e., $S=1$). On Line 39, the author mention that $O(\sqrt{\log A})$ has been improved to $O(\sqrt{\log A^{\frac{M}{A}}})$ in the expert setting. However, letting $S=1$, the work actually achieves the improvement from $O(\sqrt{\log A})$ to $O(\sqrt{\log M})$. Yet, $\sqrt{\log M}\ge \sqrt{\log A^{\frac{M}{A}}}$ for all $M\in (1,A]$. I would instead suggest the authors investigate whether significant improvements are possible in, e.g., the bandit-feedback setting.
2. **Parameter-free case**: Theorem 4.3 in this work shows that it is possible to achieve the sparsity-dependent upper bound in the parameter-free case. However, I notice that this only holds for a sufficiently large $K$ (specifically, $K\ge MT_*$). When the losses are not so “sparse”, this essentially means that $K\ge SAT_*$ is required. As such, I think this limits the applicability of the proposed algorithm.

[1] Chen et al. Minimax regret for stochastic shortest path with adversarial costs and known transition. COLT, 2021.

---

> ### Author Rebuttal · Authors · 2025-07-28
>
> We thank the reviewer for their thorough feedback. We address your comments and question below.
>
> # Weaknesses
>
> 1. Significance: Though the results are new to the literature, I am a bit concerned about the significance of the results. In particular, the authors obtain the upper bound of $\mathcal{O}(\sqrt{DKT_\star\log(T_{\star} SA)})$ in the full-information setting. Though the dependence on $M$ indeed matches the proved lower bound, the improvement from $\mathcal{O}(\sqrt{\log(SA)})$ to $\mathcal{O}(\sqrt{\log(M)})$ against previous work does not seem to be very valuable to me.
>
> - **While the improvement from $\log(SA)$ to $\log(M)$ is logarithmic, the gain is practically significant in scenarios where $SA$ is exponential in problem parameters (e.g., navigation or scheduling tasks). In such cases, $log(M)\ll log(SA)$ yields non-trivial and actionable improvements in regret.**
>
> - **Furthermore, an important aspect of our work is also showing that the gain cannot be more than logarithmic without further assumptions, thus fully characterising one aspect of optimal regret under sparsity.**
>
> - **We also emphasize the novelty of the regularizer we propose. In particular, it could be of independent interest to tackle more general sparse problems, specifically since it allows interpolation between the negative entropy and squared Euclidean norm, two common regularizers which induce contrasting behaviors on OMD.**
>
> 2. Further, I find that the improvement does not even recover it in the special case of the expert setting (i.e., $S =  1$). On Line 39, the author mention that $\mathcal{O}(\sqrt{\log(A)})$ has been improved to $\mathcal{O}(\sqrt{\log(A^{M/A})})$ in the expert setting. However, letting $S = 1$, the work actually achieves the improvement from $\mathcal{O}(\sqrt{\log(A)})$ to $\mathcal{O}(\sqrt{\log(M)})$. Yet, $\sqrt{\log(SA)} \ge \sqrt{\log A^{M/A}}$ for all $M \in (1,A]$. I would instead suggest the authors investigate whether significant improvements are possible in, e.g., the bandit-feedback setting.
>
> - **This is an interesting point. The notion of sparsity we consider is over the state-action space: $M = \max_k \sum_{(s,a) \in \Gamma} I[c_k(s,a) > 0]$. However, if we instead consider sparsity at a state-level: $M’ = \max_{k, s \in \mathcal{S}} \sum_{a \in \mathcal{A}} I [c_k(s,a) > 0] \leq A$, then using negative entropy regularisation and a cumulative loss bound [1], it is fairly straight-forward to show a regret bound of the form (ignoring log-factors) $\sqrt{K T^\star \frac{M’}{A} T^u(s_0)}$, where $T^u(s_0)$ is the hitting time of the uniform policy. This bound recovers the $M/A$ improvement of the expert setting when $S=1$. This result provides some further insights into the performance of OMD with negative entropy regularisation and that in particular issues arise when there is at least 1 state with non-sparse costs even though most other states may have sparse costs.**
>
> - **Now going back to our original notion of sparsity, we know that $M’ \leq M$. If $M \leq A$, then we can non-trivially bound $M’$ by $M$ and achieve a regret of $\sqrt{K T^\star \frac{M}{A} T^u(s_0)}$. In particular, if $S = 1$, then this is the case and we can recover the polynomial improvement of the expert setting. The reason we did not include these insights was that bounding $M’$ by $M$ can be loose and if it is known that $S=1$, then applying an experts algorithm is the more obvious choice. However, these subtleties are quite important and we will include a section on this in the appendix of the final version.**
>
> 3. Parameter-free case: Theorem 4.3 in this work shows that it is possible to achieve the sparsity-dependent upper bound in the parameter-free case. However, I notice that this only holds for a sufficiently large $K$ (specifically, “$K \ge MT$”). When the losses are not so “sparse”, this essentially means that $K \ge SAT$ is required. As such, I think this limits the applicability of the proposed algorithm.
>
> - **We do not view this assumption as restrictive or limiting the applications of the algorithms because these types of bounds are usually considered for sufficiently large $K$ (e.g. in [2], Theorem 9, they have a similar condition $K = \Omega(SA)$ in the bandit setting). This is also because the study of what is achievable for “small” K (< SA for example) corresponds to a fundamentally different high-dimensional version of the problem. In fact, it is not obvious that sub-linear regret is even achievable in this setting. Therefore we view the condition $K \ge MT_\star$ as a mild technical requirement in the parameter-free reduction, ensuring sufficient episodes per batch to learn effectively.**
>
> # Questions
>
> Q1: Could the authors please further explain the intuition on Line 223-229? I do not quite understand why “inducing an OMD update at points near the boundary of the space that can only move to new points in very close proximity” prevents the algorithms from obtaining a sparsity-dependent bound.
>
> - **Here are some more details. In what follows, we use the Euclidean distance as an alternative to measure distances but it could be other types of Bregman divergence such as the one of the regularizer we propose.**
>
> - **The main point is that the negative entropy’s gradient diverges on the boundary of the space. As a result, near the boundary of the space, the Bregman divergence of the negative entropy (the KL) between two points can be very large while those two points are very close in terms of other Bregman divergences like the Euclidean distance. If in a round, the algorithm is playing a point (in our case an occupancy measure) that is very close to the boundary (in terms of Euclidean distance), then the update of the point will be very close to the original point (again in terms of Euclidean distance) because in terms of the KL divergence it is not close. While this property can be crucial in some settings, for sparse problems this appears to be prohibitive because this is the property/intuition we use to design the MDP construction on which OMD with negative entropy regularization fails.**
>
>
>
>
> [1] Zhao, Peng, Long-Fei Li, and Zhi-Hua Zhou. "Dynamic regret of online Markov decision processes." International Conference on Machine Learning. PMLR, 2022.
>
> [2] Chen, Liyu, Haipeng Luo, and Chen-Yu Wei. "Minimax regret for stochastic shortest path with adversarial costs and known transition." Conference on Learning Theory. PMLR, 2021.

---

> > ### Comment · Reviewer_9PaL · 2025-08-04
> >
> > I appreciate the authors’ responses, which partially addressed my concerns. However, my main concern remains unresolved. I do agree that in some applications, the improvement from $\log (SA)$ to $\log M$ might be of practical interest. Nevertheless, from the theoretical perspective, I have to say that this improvement is not particularly appealing to me, though it indeed matches the lower bound. As I mentioned in my previous review, I believe investigating other problems with potential significant improvements (e.g., the bandit-feedback case) might be a more valuable starting point. Therefore, I’d like to maintain my current rating for this work.

---

> > > ### Author Response · Authors · 2025-08-05
> > >
> > > We appreciate the reviewer's response and would just like to conclude that from a theoretical standpoint, eliminating logarithmic factors (and achieving the minimax rate) is meaningful in its own right, as it often reveals deeper structural properties of the problem and calls for novel algorithmic approaches. A classic example is [1] and another is the discussion on removing extra logarithmic factors in Nemirovski-style acceleration, as noted in a blogpost by Bubeck [2]. In our case, we show that removing the extra log factor is not possible using standard OMD with negative entropy, which motivates the introduction of a new algorithmic regularizer along with a novel theoretical analysis.
> > >
> > > [1] Jean-Yves Audibert, Sébastien Bubeck. Minimax policies for adversarial and stochastic bandits. COLT, Jun 2009, Montreal, Canada. pp.217-226. ⟨hal-00834882⟩.
> > >
> > > [2] Nemirovski’s acceleration, I’m a bandit blog, Sebastian Bubeck.

---

### Decision · Program_Chairs · 2025-09-17

**Decision:**

Accept (poster)

**Comment:**

This work studies an important problem in online/reinforcement learning known as Stochastic Shortest Paths. The focus is on investigating the role of sparsity of costs in regret bounds. The main contributions include the observation that (logarithmic) scaling with the full action space is inherent to entropic mirror-descent algorithms, while a different regularizer, based on the p-norm attains bounds scaling logarithmically with sparsity. Reviewers appreciate the technical novelty of the work, but have serious concerns about its relevance: some of the assumed bounds on the instance parameter may render this algorithm inapplicable to any reasonable instance.

Despite these concerns, I believe the technical parts of this work are solid contributions. Furthermore, once the theoretical possibilities are established, follow up work can concentrate on mitigating these limitations, and turning these ideas into practical methods.

I recommend acceptance of this paper, but I encourage authors to take the following steps for their final version:
1. Include a thorough discussion about limitations and challenges towards making the algorithm practical. I think this could be a good instance to perhaps propose some interesting open problems.
2. To my knowledge, the "novel regularizer" (4) has been used for decades in convex optimization. Versions of it can be traced back to the Nemirovski-Yudin monograph, and it can be found more explicitly in recent surveys, e.g. https://www2.isye.gatech.edu/~nemirovs/ActaFinal_2013.pdf. It seems to me that this regularizer may also have connections with Tsallis entropy. My suggestion is then to put this regularization idea in perspective, and cite some relevant works.